# TK1 expression influences pathogenicity by cell cycle progression, cellular migration, and cellular survival in HCC 1806 breast cancer cells

Eliza E. Bitter[1,2], Jonathan Skidmore[1], Carolyn I. Allen[1], Rachel I. Erickson[1], Rachel M. Morris[1], Toni Mortimer[1], Audrey Meade[2], Rachel Brog[2], Tim Phares[2], Michelle Townsend[1,2], Brett E. Pickett[1], Kim L. O'Neill[1]*

1 Department of Microbiology and Molecular Biology, Brigham Young University, Provo, Utah, United States of America, 2 Thunder Biotech Inc., Provo, Utah, United States of America

* kim_oneill@byu.edu

**Data Availability Statement:** All relevant data is within the manuscript and its Supporting information files. The data provided for figures

## Abstract

Breast cancer is the most common cancer diagnosis worldwide accounting for 1 out of every 8 cancer diagnoses. The elevated expression of Thymidine Kinase 1 (TK1) is associated with more aggressive tumor grades, including breast cancer. Recent studies indicate that TK1 may be involved in cancer pathogenesis; however, its direct involvement in breast cancer has not been identified. Here, we evaluate potential pathogenic effects of elevated TK1 expression by comparing HCC 1806 to HCC 1806 TK1-knockdown cancer cells (L133). Transcriptomic profiles of HCC 1806 and L133 cells showed cell cycle progression, apoptosis, and invasion as potential pathogenic pathways affected by TK1 expression. Subsequent in-vitro studies confirmed differences between HCC 1806 and L133 cells in cell cycle phase progression, cell survival, and cell migration. Expression comparison of several factors involved in these pathogenic pathways between HCC 1806 and L133 cells identified p21 and AKT3 transcripts were significantly affected by TK1 expression. Creation of a protein-protein interaction map of TK1 and the pathogenic factors we evaluated predict that the majority of factors evaluated either directly or indirectly interact with TK1. Our findings argue that TK1 elevation directly increases HCC 1806 cell pathogenicity and is likely occurring by p21- and AKT3-mediated mechanisms to promote cell cycle arrest, cellular migration, and cellular survival.

## I. Background

Thymidine Kinase 1 (TK1) was first identified as a salvage pathway enzyme responsible for thymidine recovery during DNA synthesis [1]. Although TK1 is not essential for DNA synthesis, it is essential for cell recovery following DNA damage [2]and is tightly regulated at all levels of expression [3–7]. It is well known that abnormalities in cell proliferation and DNA repair contribute to tumor phenotype, development, and progression [8–16].

using the Cancer Cell Line Encyclopedia and The Cancer Genome Atlas (TCGA) are available from 'https://osf.io/gqrz9/' and http://timer.cistrome.org (Paper DOI: 'https://doi.org/10.1093/nar/gkaa407') respectively.

**Funding:** The author(s) received no specific funding for this work.

**Competing interests:** The authors have declared that no competing interests exist.

TK1 upregulation in neoplastic cells is an early event and has been shown to be detectable in cancer patients before clinical symptoms develop [17]. Increased expression of TK1 has been shown to be associated with cancer aggressiveness. For example, TK1 serum levels progressively increase with cancer grade (T1-T4) in breast tissue [17, 18]. Additionally, elevated serum TK1 levels in humans have been observed in lung, prostate, and colon cancer [4, 5, 18–20]. Therefore, TK1 serum expression could potentially aid in identifying malignancies across multiple cancer types [3–5, 10, 18, 19, 21, 22]. Tumor metastasis is the main cause of death among all cancer patients [23]. Breast cancer is among the most aggressive cancer types [24] with 20–30% of breast cancer cases becoming metastatic. Although some studies have suggested a potential role of TK1 in tumor development [25–27], the effects of increased TK1 expression in breast cancer pathogenesis have not been fully characterized.

In this study, we aim to explore the relationship of TK1 expression to cancer-promoting pathways involved in breast cancer pathogenesis. HCC 1806 cells were selected based on their innate aggressive nature as a primary based breast cancer cell line. To identify the potential cancer-promoting pathways that are influenced by TK1 we compared HCC 1806 wild-type and HCC 1806 TK1-knockdown (L133) cells.

## II. Methods

### 1. Comparing TK1 levels in metastatic and primary cancer cell lines

RNA-seq data was obtained through the Cancer Cell Line Encyclopedia (CCLE) [28] and classified as either "metastatic" or "primary" based on ATCC® and Cellosaurus. The expression levels of TK1 in 128 cell lines from the CCLE using data that had been generated using Illumina-based RNA-Sequencing was evaluated [28, 29]. The data values were calculated at the isoform level using the *kallisto* software [30]; the TK1 gene-level summarized values were calculated for each cell line by summing the isoform values across the six different transcripts of TK1. These values were then log-transformed to obtain transcripts-per-million (TPM) values.

To calculate significant differential expression of TK1 transcripts between primary and metastatic samples, we used a permutation-based test. Specifically, the primary/metastatic labels were permuted (n = 10,000) prior to calculating the difference in mean expression; then the actual difference in expression for TK1 was compared against its respective permuted distribution; lastly, an empirical p-value was calculated by determining the proportion of times that the actual difference was greater than the permuted differences. These tests were performed using the R (v.3.4.3) statistical software.

### 2. Identifying upregulated levels of TK1 across cancer types

Differential TK1 expression was evaluated between adjacent normal and tumor tissue samples for 39 different cancer types using RNA-seq data extracted from TCGA with the TIMER 2.0 program [31–33] (S1 Fig). Files were obtained for "primary solid tissue" and for "solid normal tissue" for each cancer type and the estimation of generated transcripts (scaled estimates) were transformed to TPM. The TPM for TK1 expression comparing normal tissue and tumor tissue were represented by box and whisker plots which depict the 25th percentile, median, 75th percentile and maximum values. Statistical significance was computed using a Wilcoxon test.

### 3. Evaluating the methylation level of the TK1 promoter in normal and primary tumor samples using TCGA data

The promoter DNA methylation level in TCGA breast cancer samples [34, 35] was determined by using bisulfite converted DNA from human invasive-breast cancer cells and normal cells

that were analyzed through Illumina Infinium HumanMethylation450 BeadChip array (HM450K, Illumina Inc). Nine Illumina ID groups were used to assess the methylation of the transcriptional start site (S2 Fig). Five of these were 0–200 base pairs upstream of the transcriptional start site (TSS200) and four of these were 200–1500 base pairs upstream of the TSS (TSS1500). The β values, whichrepresent the ratio of intensities between methylated and unmethylated signals, were calculated using the following formula:

$$\beta = \text{signal\_methylation} \div (\text{signal\_methylation} + \text{signal\_unmethylation})$$

The β value indicates the level of DNA methylation ranging from 0 (fully unmethylated) to 1 (fully methylated). The significant difference between the tumor and normal samples was estimated by a student's t-test (unequal variance was considered).

## 4. CRISPR-Cas9 preparation of HCC 1806 TK1 knockdown cell line L133

CRISPR plasmid pSpCas9(BB)-2A-Puro (PX459) V2.04 was acquired from Addgene. The plasmid was handled according to standard procedures. Guide RNAs (sgRNAs) were designed using suggestions provided by Addgene as well as online scoring sources such as tefor CRISPOR, CRISPR MIT, and CRISPR RGEN Tools [36–40].

HCC 1806 cells were seeded at a concentration of $5 \times 10^5$ cells in each well of a six-well tissue-culture treated plate and incubated overnight prior to transfecting PX459. Lipofectamine® LTX DNA Transfection Reagent (Thermo Scientific™, 75 Panorama Creek Drive Rochester, NY 14625 USA) was used according to the manufacturer's instructions for cell transfection. LTX reagent was diluted into Opti-MEM™ to a 6% concentration. In a separate tube, DNA (9 ng) was diluted in PLUSTM reagent in a 1:1 ratio and added to 450 μl of Opti-MEM™. Both tubes were incubated separately for five minutes prior to adding the solution containing DNA to the LTX reagent in Opti-MEM™. Once combined, they were incubated together for 20 minutes. Treatment was added dropwise to corresponding wells for a final concentration of 3 ng PX459. Cells were incubated for three days at 37° C and 5% $CO_2$. HCC 1806 TK1-knockdown (L133) transfected cells were selected with puromycin (2ug/ml) until control cells were eliminated. Stable knockdown of TK1 was confirmed through Western Blotting and qRT-PCR.

## 5. Cell lysate preparation and Western Blot quantification

To obtain lysates, cells were grown to 100% confluency in a T150 flask and trypsinized prior to treatment with ~1.5 ml of NP40 lysis buffer from Invitrogen™ supplemented with protease inhibitor cocktail for 30 minutes on ice and vortexing every 10 minutes. Cell lysate samples were aliquoted and stored at -80°C. The Pierce™ BCA Protein Assay Kit from Thermo Scientific™ was used to quantify protein concentration of cell lysate samples according to manufacturer's instructions. Protein concentration was analyzed on a BioTek® plate reader at 562nm. Western blot analysis was performed using ~30 μg of cell lysate from respective samples. Samples were thawed on ice and prepared with SDS loading dye and water before being boiled for 5 minutes, with ~30 μg of protein loaded per well. The samples were loaded into a 12% acrylamide SDS gel and run at 90 V for 2–3 hours. Proteins were transferred onto a nitrocellulose membrane at 90 V for 50 minutes. The nitrocellulose membrane was blocked with 5% nonfat milk in PBS for 1 hour at room temperature (RT) on a rotating platform. All further washes and incubations were carried out on a rocking platform. Samples were washed in PBS at RT and were incubated with either anti-TK1 commercial antibody (Abcam® 76495) or anti-GAPDH antibody (Cell Signaling® 14C10). All primary antibodies were incubated in a 1:1,000 dilution in PBS overnight at 4°. The membrane was washed three times in PBS at RT and incubated with an HRP-goat anti-rabbit secondary antibody for 1 hour at 4°C prior to

imaging. The membrane was washed three times with PBS and exposed following manufacturer's instructions for Advansta WesternBright™ ECL horseradish peroxidase substrate. Western Blots were then exposed to film and developed. Protein size was determined by comparing to ExcelBand™ All Blue Broad Range Protein Marker from SMOBIO®.

Samples were tested in biological triplicates and were quantified using ImageJ software [41] to calculate the protein band intensity. Samples were referenced to GAPDH control for relative intensity. Difference in intensity was assessed by a student's t-test in GraphPad PRISM®.

## 6. Quantification of TK1 through qRT-PCR

Difference in RNA transcript expression was compared between HCC 1806 and L133 cells through quantitative real-time polymerase chain reaction (qRT-PCR). RNA was isolated from HCC 1806 and L133 using Fisher BioReagents™ SurePrep™ TrueTotal™ RNA Purification Kit (catalog: BP 2800–50). Approximately $2 \times 10^6$ cells were first collected using trypsin and washed twice with 1x PBS. The samples were then treated with cell lysate, mixed with 95% ethanol, and centrifuged for 10 minutes at 200 x $g$. The supernatant was then transferred to a spin column assembly and centrifuged for one minute. The column was washed twice with wash buffer and spun at 200 x g for one minute and then a final time for two minutes. Following the washes, the column was transferred to a clean centrifuge tube and eluted with molecular grade water and centrifuged for two minutes at 200 x $g$ and one minute at 14,000 x $g$. RNA purity and concentration was determined by gel electrophoresis and absorbance (A) ratio $A_{260}/A_{280}$. Absorbance was measured using a NanoDrop™ One/OneC Microvolume UV-Vis Spectrophotometer (Thermo Scientific™, 701–058112).

Following RNA isolation, samples were converted to cDNA using GoScript™ Reverse Transcription System (Promega, #A5000) according to the manufacturer's protocol. Briefly, extracted RNA was mixed with Oligo(dT)$_{15}$ primer and nuclease free water. This mixture was heated to 70˚C for 5 minutes and then chilled on ice-water for 5 minutes. In a separate tube, GoScript™ Reverse Transcriptase was prepared with reaction buffer, MgCl$_2$, PCR nucleotide mix, RNase inhibitor, and water. Approximately 15 μl of this mixture was added to each primed RNA sample. Experimental samples were then annealed for 5 minutes at 25˚C and extended for 1 hour at 42˚C. An additional step for reverse transcriptase was included at 70˚C for 15 minutes. cDNA samples were assessed for concentration and purity (260/280 $\geq$ 1.8) using a Nanodrop instrument.

Prepared cDNA from HCC 1806 and L133 cells was then analyzed for TK1 expression through qRT-PCR. Target (TK1) and reference (GAPDH) samples for HCC 1806 (n = 6) and L133 (n = 6) cDNA was then prepared in a MicroAmp™ Fast Optical 48-Well Reaction Plate (Applied Biosystems™). To each well, SYBR™ green PCR master mix (Applied Biosystems™), 100 ng of cDNA, validated primers (Integrated DNA Technologies, GAPDH # Hs.PT.39a.22214836, TK1 #Hs.PT.58.4323868), and water was added. The following is the primer set for TK1, reverse: CGTCTCATCAACTCTGTGCTTT and forward: GCATTAACCTGCCCACTGT. A MicroAmp™ Optical Adhesive Film (Applied Biosystems™) was used to seal off samples prior to analysis on a StepOnePlus™ Real-Time PCR System (Applied Biosystems™). The $C_T$ values for each replicate (n = 3) were subsequently used to calculate the relative quantity of TK1 (RQ = $2^{-\Delta\Delta CT}$). The relative quantity of TK1 in HCC 1806 and L133 samples was tested for a significant difference using a student's t-test in PRISM® (GraphPad Software, Inc.).

## 7. Nitrophenyl phosphate cell growth assay

Standard growth curves were generated for each cell line. Cells were grown in 96-well plates at seeding densities from 0 to 1.2 x$10^4$ cells per well, increasing by increments of 1x$10^3$cells. For each seeding cell density, samples were plated in six replicates. The culture medium was

removed, and the cells were washed once with PBS. To each well, 100 μl of buffer containing 0.1 M sodium acetate (pH 5.0), 0.1% Triton 100X, and 5mM p-nitrophenyl phosphate was added. The plates were placed in a 37˚C incubator for two hours. The reaction was stopped with the addition of 20 μl of 1 M NaOH, and color development was assessed at 405 nm on a Biotek [R] Synergy™ HT microplate reader.

To estimate the average frequency (number of cell divisions per day), the same number of cells were plated (~3000) in replicate (n = 6). To each well, 100 μl of buffer containing 0.1 M sodium acetate (pH 5.0), 0.1% Triton 100X, and 5mM p-nitrophenyl phosphate was added. Following 24 hours, the reaction was stopped with the addition of 20 μl of 1 M NaOH, and color development was assessed at 405 nm on a Biotek [R] Synergy™ HT microplate reader. Individual standard curves for HCC 1806 and L133 cells were then used to determine the cell count based on color development readings. From here, we were able to calculate the average cell division per day for both cell types. Significant difference in average frequency was assessed in PRISM[R] (GraphPad Software, Inc.) using a student's t-test.

## 8. RNA-seq study of HCC 1806 and L133 cells

**8.1 RNA extraction and purification.** RNA was isolated using SV Total RNA Isolation System (Promega, #Z3105) according to the manufacturer's protocol. In brief, cells were lysed in RNA lysis buffer containing beta-mercaptoethanol and needle shearing was used to physically reduce the size of the genomic DNA. The samples were then diluted and centrifuged. The cleared lysate was transferred to a new tube and incubated in 95% ethanol. This mixture was moved to a spin column assembly and centrifuged. In preparation for RNA isolation, each sample was incubated in a DNase mixture containing buffer, $MnCl_2$, and DNase for 15 minutes at room temperature. The reaction was stopped using a DNase stop solution and washed once with wash buffer. Samples were eluted in nuclease free water and assessed for purity (260/280 $\geq$ 1.8) and concentration using a NanoDrop instrument (Thermo Fisher Scientific). Immediately following isolation, samples were moved to -80˚C storage. Completed samples were sent to LC Sciences (2575 W Bellfort Ave Ste 270, Houston, TX 77054) on dry ice for RNA-seq processing and analysis. The quantity, purity, and integrity of samples were assessed by LC Sciences. RNA sample integrity was evaluated using an Agilent Bioanalyzer 2100.

**8.2 RNA-seq library construction and sequencing.** Poly(A) RNA sequencing libraries were prepared following the Illumina TruSeq stranded-mRNA sample preparation protocol. RNA integrity was checked with Agilent Technologies 2100 Bioanalyzer. Poly(A) tail-containing mRNAs were purified using oligo-(dT) magnetic beads with two rounds of purification. After purification, poly(A) RNA was fragmented using divalent cation buffer at elevated temperature. Quality control analysis and quantification of the sequencing libraries were performed using Agilent Technologies 2100 Bioanalyzer High Sensitivity DNA Chip. Paired-ended sequencing was performed on Illumina's NovaSeq 6000 sequencing system.

**8.3 RNA-seq processing and analysis.** To remove the sequencing adapters, low-quality regions of reads that contained adaptor contamination, and ambiguous bases, Cutadapt [42] was initially applied. Sequence read quality was then verified using FastQC (http://www.bioinformatics.babraham.ac.uk/projects/fastqc/). HISAT2 [43] was used to map reads to the human genome of (Ensembl release 96; ftp://ftp.ensembl.org/pub/release-96/fasta/homo_sapiens/dna/). The mapped reads of each sample were compiled using StringTie [44]. All transcriptomes were then merged to reconstruct a comprehensive transcriptome using Perl scripts and gffcompare. After the final transcriptome was generated, StringTie [44] was used to quantify the expression levels of all transcripts. The differentially expressed mRNAs were calculated

with edgeR, followed by applying the following selection criteria: log2 (fold change) > |1| and statistical significance (p value < 0.05) [45].

## 9. Cell cycle analysis for L133 and HCC 1806 cells

To detect differences in cell cycle phases of HCC 1806 and L133 cells, a total of $3.5 \times 10^5$ cells were plated in each well of a six well plate 24 hours prior to cell cycle analysis. Once samples were in exponential phase, they were trypsinized and washed twice with PBS prior to fixation in 4% paraformaldehyde. Following fixation, samples were washed once with 1x PBS and placed in 70% methanol overnight. All samples were washed a final time in 1x PBS and treated with RNase. Test samples were treated with propidium iodide for 30 minutes at 4˚C. Positive staining for propidium iodide was detected on a Beckman Coulter Cytoflex $^{®}$ and cell cycle analysis was performed in FlowJo, LLC Cell Cycle Platform software$^{®}$. To test for a significant difference within each phase of the cell cycle between HCC 1806 and L133 samples, a student's t-test was used in PRISM$^{®}$ (GraphPad Software, Inc.).

## 10. Determining survival of L133 and HCC 1806 cells under serum deprivation and hypoxic conditions through apoptosis

HCC 1806 and L133 cell lines were plated in triplicate in 6 well plates with $5 \times 10^5$ cells per well. Cells were allowed to adhere to the plates for ~2–3 hours before being placed under glucose deprivation or hypoxic stress. Hypoxic stress was induced with Cobalt (II) Chloride hexahydrate ($CoCl_2$ 6H2O) at a final concentration of 50 μM. The molecular use of $CoCl_2$ 6H2O in cell culture induces a hypoxic environment [46]. Serum starvation was induced with a mixture of cell-culture media + 2% fetal bovine serum (FBS). Both conditions were evaluated at 12 and 30 hours and repeated three times.

Early apoptotic cells were identified through phosphatidylserine translocation to the outer membrane in a compromised cell. The translocation event was measured by Annexin V binding using the BD Pharmigen™, FITC Annexin V Apoptosis Detection Kit I. Later stages of apoptosis and necrotic cells were detected by a combination of Annexin V and propidium iodide (PI) staining [47–49]. The tested samples were examined for Annexin V and/or PI binding on a Beckman Coulter Cytoflex $^{®}$ and results were analyzed using FlowJo $^{®}$ software. Within each cell line, differences between sample and control means were tested for using a one-way ANOVA test. For differences in population of phosphatidylserine between HCC 1806 and L133 cells, a student's t-test was used.

## 11. Evaluating invasion of L133 and HCC 1806 cells by scratch assays

The following method was adapted from Pinto et al. [50]. Briefly, L133 and HCC 1806 cells were seeded with ~$3.5 \times 10^5$ cells in a 12 well plate to 100% confluency. Media was removed and a 100 μl pipette tip was used to make a scratch down the center of each well. Wells were washed with 1x PBS and 4 mL of 5% FBS RPMI-1640 media were placed in each well. Immediately, prepared cells were taken to be imaged on an ImageXpress$^{®}$ Pico Automated Cell Imaging System (Molecular Devices) at 20x magnification every hour until gap closure (< 24 hours). An environmental control to maintain 5% C0$_2$, 37˚C, and ~80% relative humidity was used during imaging. All scratches closed prior to 24 hours and each individual scratch was normalized for analysis. For differences in slopes between HCC 1806 and L133 cells, a slope comparison test was used.

## 12. QRT-PCR fold change for cell cycle checkpoint, cell adhesion, and apoptotic factors

Differences in RNA transcript expression was compared between HCC 1806 and L133 cells through qRT-PCR. RNA was isolated from cells using Invitrogen™ PureLink™ RNA Mini Kit (catalog: Invitrogen™ 12183018A) using manufacturer's protocol. In brief, approximately 3 x $10^6$ cells were first collected with trypsin. Samples were then treated with cell lysis buffer and mixed with 70% ethanol. The sample was transferred to a spin cartridge and centrifuged. The column was washed twice with a series of wash buffers and then a final spin to remove any residual buffer. The column was then transferred to a clean centrifuge tube and eluted with molecular grade water. RNA purity and concentration was determined by gel electrophoresis and absorbance (A) ratio $A_{260}/A_{280}$ using a NanoDrop™ One/OneC Microvolume UV-Vis Spectrophotometer (Thermo Scientific™, 701–058112).

Following RNA isolation, samples were converted to cDNA using GoScript™ Reverse Transcription System (Promega, #A5000) according to manufacturer's protocol. In short, experimental RNA was mixed with Oligo(dT)$_{15}$ primer and nuclease free water. This mixture was heated to 70˚C for 5 minutes and then chilled on ice-water for 5 minutes. In a separate tube, GoScript™ Reverse Transcriptase was prepared with reaction buffer, $MgCl_2$, PCR nucleotide mix, RNasin inhibitor, and water. Approximately 15 µl of this mixture was added to each primed RNA sample. Experimental samples were then annealed for 5 minutes at 25˚C and extended for 1 hour at 42˚C. An additional step for reverse transcriptase was included at 70˚C for 15 minutes. cDNA samples were assessed for concentration and purity ($260/280 \geq 1.8$) using a Nanodrop instrument.

Prepared cDNA was then analyzed for expression of the respective factor (Table 1) through qRT-PCR. Target and reference (GAPDH) samples for HCC 1806 and L133 cDNA was then prepared in a MicroAmp™ Fast Optical 48-Well Reaction Plate (Applied Biosystems™). To each well, SYBR™ green PCR master mix (Applied Biosystems™), 100 ng of cDNA, validated primers (Table 1), and water was added. A MicroAmp™ Optical Adhesive Film (Applied Biosystems™) was used to seal off samples prior to analysis on a StepOnePlus™ Real-Time PCR System (Applied Biosystems™). The $C_T$ values for each sample were subsequently used to calculate the relative quantity of TK1 ($RQ = 2^{-\Delta\Delta CT}$). The fold changes shown for L133 cells is relative to the expression of HCC 1806.

## 13. Examining TK1 correlation to cell adhesion, apoptotic, and cell cycle checkpoint factors in BRCA patients

The online bioinformatics tool TIMER, developed by Li T., et al. [31], was used to evaluate correlations between TK1 and invasive, apoptotic, or cell cycle checkpoint factors using RNA-seq BRCA patient data (n = 1093). Contributing factors for invasive, apoptotic, and cell cycle checkpoint pathways were gathered using the PathCard and GeneCard databases [51, 52]. The degree of correlation was represented by purity-adjusted partial spearman's rho value (S3–S7 Figs).

## 14. Predicting TK1 interactions with cell cycle checkpoint, invasion, and apoptotic factors

STRING Database Software was used to predict protein-protein interactions with cell cycle checkpoint, invasion, and apoptotic factors evaluated by qRT-PCR analysis [53]. A total of 24 factors were evaluated for potential associations to TK1. In short, the STRING Database collects and scores evidence based on text mining using the following methods: 1) Automated

**Table 1. Factors tested in qRT-PCR expression analysis with the validated primer catalog ID from IDT and their molecular function based on STRING database.**

| Factor | Validated Primer Catalog ID | Molecular Function |
|---|---|---|
| GAPDH | Hs.PT.39a.22214836 | Positive control |
| AKT3 | Hs.PT.58.2121896 | Apoptosis |
| ATM | Hs.PT.56a.2596352 | Apoptosis + Cell Cycle |
| BRCA1 | Hs.PT.56a.27724517.g | Apoptosis + Cell Cycle |
| CCNY | Hs.PT.58.25126620 | Apoptosis + Cell Cycle |
| CDH18 | Hs.PT.58.3054591 | Invasion |
| CHEK2 | Hs.PT.58.896970 | Apoptosis + Cell Cycle |
| CLOCK | Hs.PT.58.20568760 | Cell Cycle |
| E2F1 | Hs.PT.58.45513742 | Apoptosis + Cell Cycle |
| FHL2 | Hs.PT.58.27184382 | Apoptosis |
| FLNB | Hs.PT.58.15270439 | Invasion |
| GAS6 | Hs.PT.58.21535693 | Apoptosis + Invasion + Cell Cycle |
| MAD1L1 | Hs.PT.58.26024703 | Cell Cycle |
| MDM2 | Hs.PT.58.358457 | Apoptosis + Cell Cycle |
| NFKB | Hs.PT.58.40427794 | Apoptosis |
| P21 | Hs.PT.58.23033764 | Apoptosis + Cell Cycle |
| PIK3R3 | Hs.PT.58.5066731 | Apoptosis + Invasion |
| PPP2R2B | Hs.PT.58.2153902 | Apoptosis + Cell Cycle |
| PRMT2 | Hs.PT.58.21184168 | Cell Cycle |
| RAD52 | Hs.PT.58.40737446 | Apoptosis + Cell Cycle |
| RB1 | Hs.PT.58.785814 | Apoptosis + Cell Cycle |
| TICRR | Hs.PT.58.24717487 | Apoptosis + Cell Cycle |
| TOPBP1 | Hs.PT.58.1879430 | Apoptosis + Cell Cycle |
| TP53 | Hs.PT.58.39676686 | Apoptosis + Cell Cycle |

text mining of the scientific literature, 2) Databases of interaction experiments and annotated complexes/pathways, 3) Computational interaction predictions from co-expression and from conserved genomic context, and 4) Systematic transfers of interaction evidence from one organism to another. The information from STRING Database was retrieved and modified for improved readability.

## 15. Immunohistochemistry of invasive ductal and infiltrating lobular tissues

A breast cancer tissue array (BRM961a) was obtained from US Biomax, Inc. for evaluating the staining intensity of breast cancer tissue of TK1. The histology slide contained 48 cases of breast carcinoma, 36 cases of metastatic carcinoma, and 12 cases of matched cancer adjacent or adjacent normal breast tissue. The breakdown of the samples used in our analysis are as follows: adjacent normal breast (n = 8), infiltrating lobular (n = 7), invasive ductal (n = 39), metastatic ductal (n = 30) and metastatic lobular (n = 6). The controls used in TK1 level analysis were GAPDH and a universal human isotype.

Tissues were stained according to protocols outlined in Townsend et al. [54]. Briefly, slides were rehydrated in decreasing amounts of alcohol prior to DIVA (BioCare medical, LLC) treatment and blocked for non-specific binding. Tissues were incubated for 12–16 hours in a humidity chamber at 4°C in primary antibody, either GAPDH (Cell Signaling®, address, country; clone 14C10) or anti-TK1 (Abcam®; 1 Kendall Square, Suite B2304 Cambridge, MA

02139–1517, USA; clone 76495). Tissues were washed in Tris-buffered saline (TBS) three times treated with secondary anti-rabbit HRP (MACH 4 HRP polymer, Biocare Medical,LLC) and incubated at room temperature for 30 minutes in a humidity chamber. Tissues were washed in TBS three times to remove unbound secondary antibody and then exposed to 3,3′-Diaminobenzidine (DAB). Slides were then treated with hematoxylin, washed in cold running water and dehydrated in increasing amounts on alcohol before being mounted using Cysto-seal™ 280 (Thermo Scientific™).

Following tissue imaging, images were analyzed using ImageJ software [41]. Briefly images were filtered for DAB staining using 'IHC toolbox' program with a selected "more DAB" option [41] and then converted to a gray scale and placed under a set threshold based on positive and negative controls (GAPDH and universal human isotype). A threshold of 50–150 was applied to avoid bias for negative space within the image. Once the threshold was applied to all images, each image was assessed for average gray intensity. Quantified tissues were then analyzed in PRISM® (GraphPad Software, Inc.) using a multiple comparison ANOVA test to compare TK1 levels in breast tissue type to positive and negative controls.

## 16. Statistical analysis

All data analysis was performed using GraphPad Prism (Version 5.0; GraphPad Software, La Jolla, USA). Data are presented as the standard error of mean. Significant differences between groups were evaluated using Student's t test, slope comparison test, one-way ANOVA, or two-way ANOVA. The confidence level for statistically significant results was 95%. Statistical significance is annotated by the number of asterisks present (*: p-value < 0.05; **: p-value < 0.01; ***: p-value < 0.001; ****: p-value < 0.0001).

## III. Results

### 1. TK1 cytosolic expression in metastatic and primary breast cancer cell lines

We first confirmed the upregulation of cytosolic TK1 in several different cancer types using RNA-seq data from The Cancer Genome Atlas (TCGA). Our analysis showed that TK1 expression in patient tissue samples across multiple cancer types was significantly higher in tumor samples as compared to healthy tissues. Specifically, 20 out of the 39 cancer types tested showed significant TK1 elevation in tumor tissue when compared to adjacent healthy tissue (S1 Fig).

We hypothesized that TK1 levels may change with disease progression and specifically that TK1 expression would be higher in metastatic cell lines as compared to primary cell lines. To test this hypothesis, RNA-seq data was obtained through the Cancer Cell Line Encyclopedia (CCLE) for 128 different cell lines across different cancer types [28]. The cell lines (n = 128) were categorized as metastatic or primary based on ATCC® and Cellosaurus classification. The TK1 levels within each cell line were computed by summing the isoform values. A permutation-based test was used to test the difference in means between primary and metastatic samples. Our results showed that metastatic cell lines contained overall higher levels of TK1 compared to primary cell lines (p <0.0001) (Fig 1).

After evaluating levels of TK1 indifferent cancer types, we wanted to narrow our research to a particular cancer type. Inherently, breast cancer is both aggressive and complex—with several derivatives or types. We predicted that TK1 levels would be higher in metastatic cell lines when compared to primary breast cell lines. Therefore, we selected several breast cancer cell lines and performed Western Blots to measure TK1 levels. These cell lines were categorized as

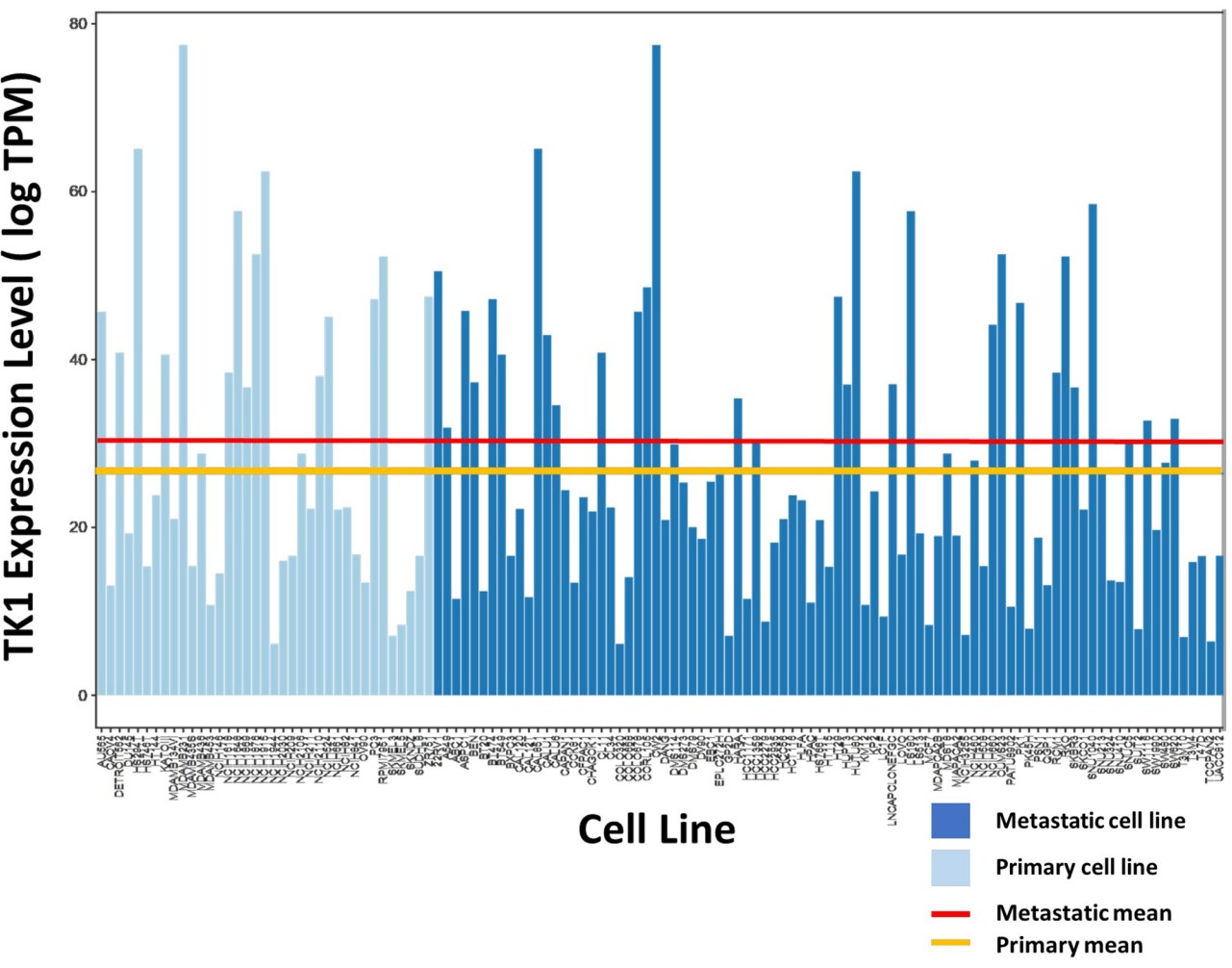

**Fig 1. Comparing RNA TK1 levels in metastatic and primary cell lines.** RNA-seq data from Cancer Cell Line Encyclopedia (CCLE) for primary and metastatic cell lines were analyzed for TK1 expression. Data was mined from a genomic RNA-seq database for well characterized cell lines. A permutation-based test was used to evaluate significance. Metastatic cell lines showed a significantly higher level for cytosolic TK1 than primary cell lines ($p < 0.0001$).

either "metastatic" or "primary" based on available literature. Our Western Blots indicated that metastatic breast cancer cell lines contained higher levels of TK1 when compared to primary breast cancer cell lines ($p < 0.05$) (Fig 2).

## 2. Exploring TK1 elevated expression via promoter dysregulation

While there are many aspects to protein regulation within a cell, we anticipated that promoter dysregulation was a contributing factor of elevated levels of cytosolic TK1. Consequently, we compared the DNA methylation levels for the TK1 promoter between invasive-breast cancer cells and healthy cells [34, 35]. Normal patient samples showed a higher level of methylation in the TK1 promoter when compared to primary BRCA tumor samples ($p = 1.11 \times 10^{-16}$) (S2 Fig). Our results suggest that promoter dysregulation contributes to elevated levels of TK1.

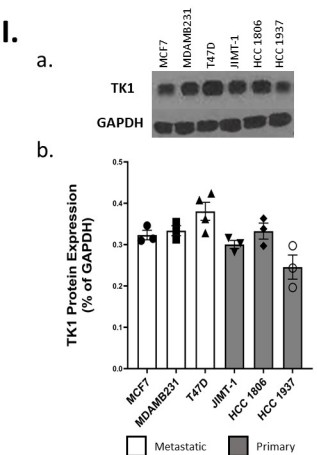
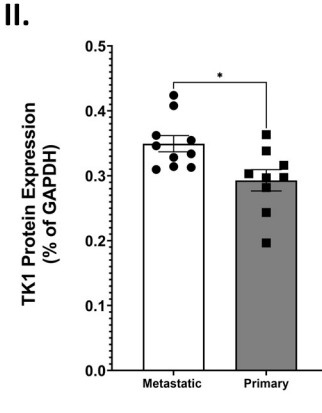

**Fig 2. Comparing TK1 protein levels in metastatic and primary breast cell lines. I. a)** Western blot analysis of cytosolic TK1 levels was performed in various types of breast cancer cell lines. **b)** Individual monomeric levels of TK1 were assessed in six different breast cancer cell lines. **II.** Cells were categorized as metastatic or primary based on current literature. A student's t-test was performed comparing TK1 levels between metastatic and primary cells. Metastatic cell lines contained a higher level of cytosolic TK1 ($p < 0.05$).

## 3. RNA-seq analysis comparing gene expression profiles of TK1 knockdown L133 cells and wild-type HCC 1806 cells

Next, we wanted to investigate if TK1 was influencing cancer-promoting pathways. We predicted that reduced TK1 levels may lead to a less aggressive phenotype when compared to elevated TK1 levels. To evaluate this, we used HCC 1806 cells, an aggressive triple-negative primary breast cancer cell line and created a TK1-knockdown HCC 1806 cell line (L133) to collect RNA-seq data to identify differences between the two cell lines in gene and transcript profiles, GO terms, and KEGG pathways.

We confirmed TK1 knockdown in the HCC 1806 cells through qRT-PCR and Western Blotting and will be referred to hereafter as L133 (Fig 3). To confirm that the TK1 knockdown did not largely affect cellular growth and bias downstream results, we performed a nitrophenyl phosphate cell growth assay. We observed that the mean average frequency (doubling time in 24 hours) differed between the HCC-1806 and L133 cells (Fig 3). While HCC 1806 cells did show a higher average frequency, we concluded that this difference was biologically minimal and would not contribute to biased differences in future experiments. In preparation for generating RNA-seq data, we isolated poly-A RNA from biological replicates (n = 3) of the HCC 1806 and L133 cells. Samples that met purity criteria ($260/280 \geq 1.8$) were submitted for gene expression quantification with RNA-sequencing.

The Gene ontology (GO) analysis of differentially expressed genes identified cell adhesion as a significant pathway linked to TK1 expression. Similarly, KEGG enrichment analysis identified cell adhesion molecules as a significant pathway. GO analysis also identified theapoptosis process as a significant GO term and additional terms relating to the cell cycle (Fig 4). Some of those cell cycle terms include variations of regulation of transcription, positive regulation of cell population proliferation, and positive regulation of gene expression. We also found that alcoholism and systemic lupus erythematosus were also significantly enriched signaling pathways. Through differential expression analysis for transcripts and genes, we observed 99 transcripts and 99 genes that significantly differed between L133 and HCC 1806 samples (Figs 5 and 6). Several notable proteins involved in apoptosis, cell adhesion, and cell cycle were

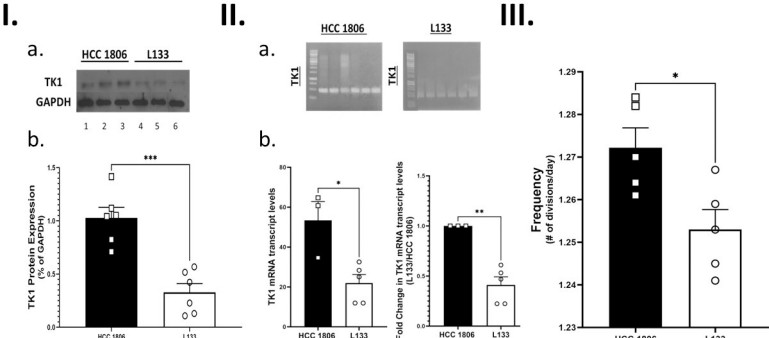

**Fig 3. CRISPR-Cas9 TK1 knockdown confirmation. I. a)** Western Blot detection for TK1 and loading control (GAPDH) in HCC 1806 and L133 cell lysate. Lanes 1–3 contain cell lysate from HCC 1806 cells and lanes 4–6 contain cell lysate from L133 cells. **b)** Difference in mean relative expression for TK1 in HCC 1806 and L133 cells was evaluated by a student's t-test in PRISM. L133 cells showed less expression of TK1 when compared to HCC 1806 cells ($p < 0.001$). **II. a)** PCR amplification of TK1 (lanes 1–6) from HCC 1806 and L133 cDNA. GAPDH was used as the reference sample (not shown). **b)** Fold change was calculated from qRT-PCR data for TK1 in HCC 1806 and L133 cells using GAPDH as the reference gene. L133 cells had a lower fold change for TK1 when compared to HCC 1806 cells ($p < 0.01$). **III.** A nitrophenyl growth assay was used to determine frequency (# of divisions/day) of L133 and HCC 1806 cells. Individual standard curves for HCC 1806 and L133 cells were used to determine the cell count based on color development readings over a 24-hour period. A student's t-test was used to compare the difference in means. L133 cells had a lower frequency when compared to HCC 1806 cells ($p < 0.05$).

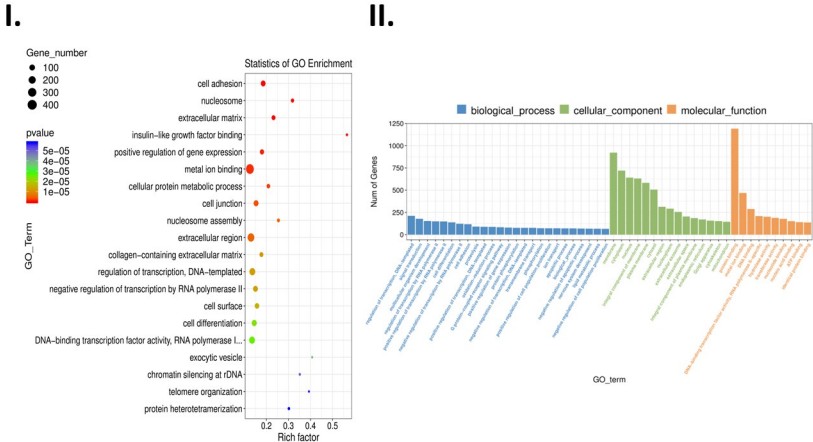

**Fig 4. Gene ontology enrichment analysis of gene product attributes from RNA-seq analysis of HCC 1806 and L133 cells. I.** Scatterplot of GO terms and the associated statistics from the GO Enrichment analysis. The legend associated with the scatterplot identifies gene number by circle size and p value by color intensity. Metal ion binding and extracellular region are some of the most prevalent GO terms. **II.** Bar plot of enriched GO terms shows number of genes associated to a given term and which system classification they belong. The legend above the graph shows the color associated with each system classification.

among those differentially expressed. Among them were phosphatase 2 (PPP2R2B), Four and a half LIM domains protein 2 (FHL2), Growth arrest—specific 6 (GAS6), Protein Kinase B Gamma (AKT3), Phosphoinositide-3-Kinase Regulatory Subunit 3 (PIK3R3), Filamin B (FLNB), Cadherin 18 (CDH18), and Protein Arginine Methyltransferase 2 (PRMT2). Thus, we suspected elevated levels of TK1 were influencing the pathogenicity of HCC 1806 cells through cell cycle progression, cell survival, and cell migration.

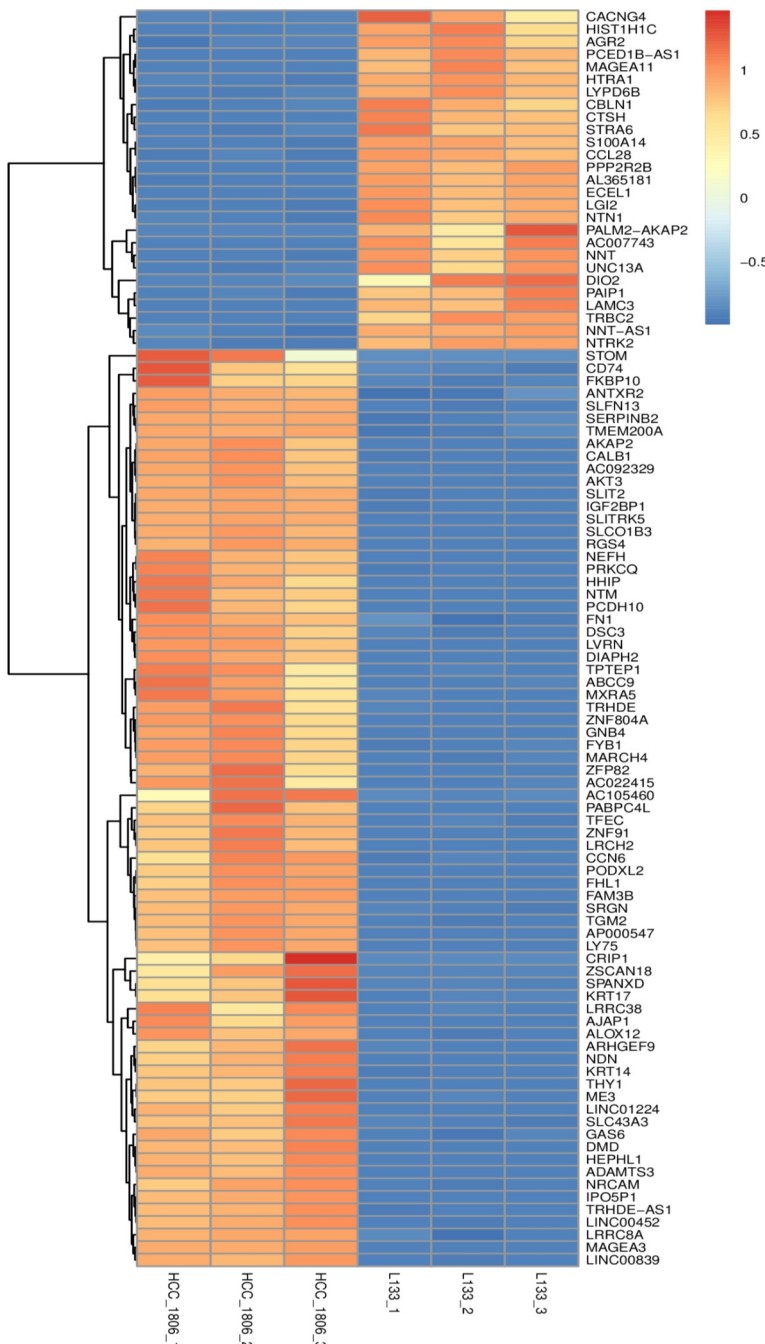

**Fig 5. Heatmap of differentially expressed genes comparing HCC 1806 and L133 cells.** Samples were tested in biological replicates (n = 3). Higher expressed factors are depicted by increasing red intensity where lower expression is depicted by increasing intensity of blue.

## 4. TK1 and its suggested role in cell cycle progression

Based on results from our RNA-seq study, and because of the direct role of TK1 in DNA synthesis, we aimed to determine whether TK1 levels were promoting cancer via a cell cycle checkpoint pathway. To observe possible changes in cell cycle phases between L133 and HCC

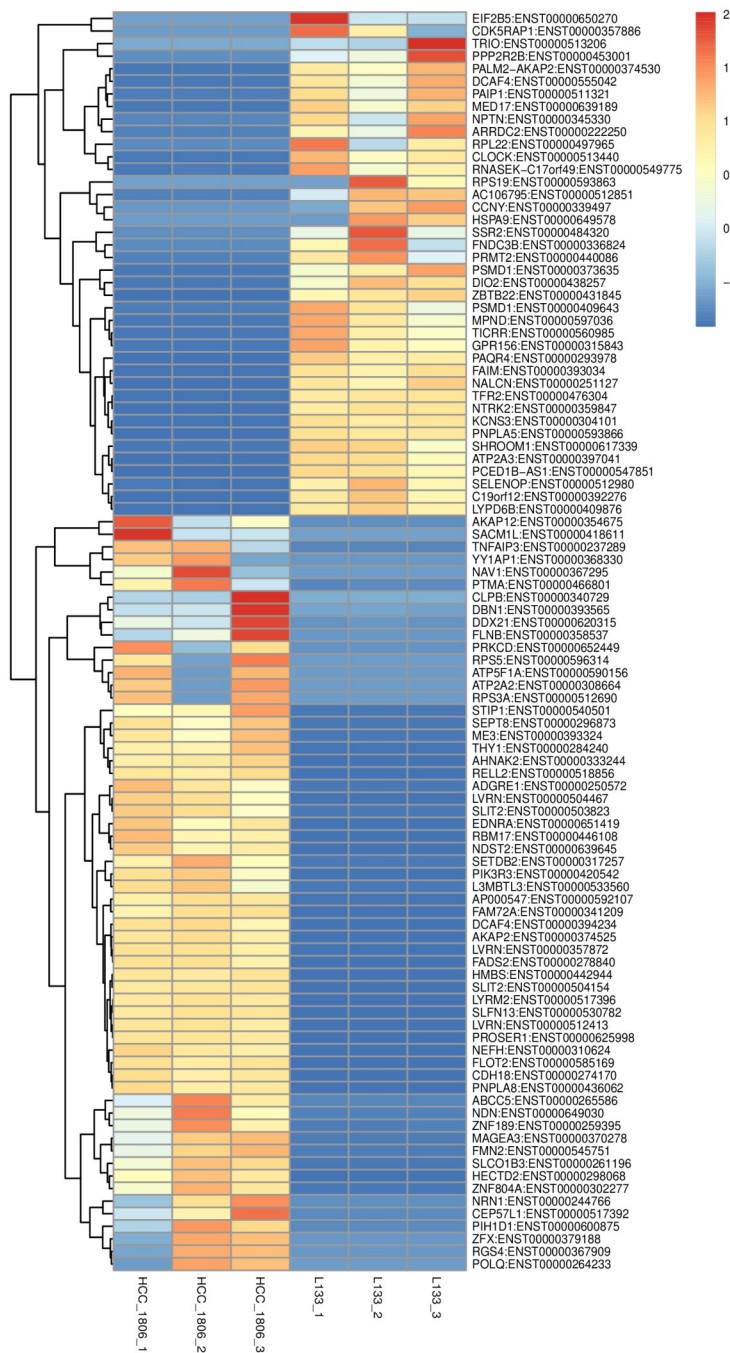

**Fig 6. Heatmap of differentially expressed transcripts comparing HCC 1806 and L133 cells.** Samples were tested in biological replicates (n = 3). Higher expressed factors are depicted by increasing red intensity where lower expression is depicted by increasing intensity of blue.

1806 cells, we performed cell cycle analysis. This analysis revealed a higher population of L133 cells in S phase compared to HCC 1806 cells (p<0.0001) and a lower G1 phase in L133 cells compared to HCC 1806 cells (p<0.0001) (Fig 7). These findings show that TK1 was indeed influencing cell cycle progression.

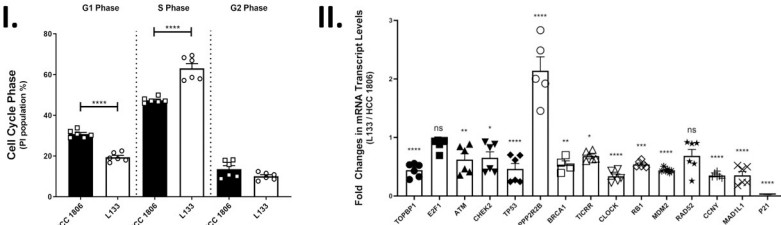

**Fig 7. The influence of TK1 on cell cycle progression, pathway, and its regulation. I.** HCC 1806 and L133 cell cycle analysis. Cell cycle progression was detected using propidium iodide staining and measured using flow cytometry. Quantification in each phase of the cell cycle was determined using the cell cycle analysis platform in FlowJo. A student's t-test was used to compare mean differences between HCC 1806 (n = 3) and L133 (n = 3) cells within each cell cycle phase. L133 had significantly more cells in S phase and significantly less cells in G1 when compared to HCC 1806 cells (p < 0.0001). **II.** Relative transcript levels in HCC 1806 and L133 cells were quantified using qRT-PCR for a subset of factors involved in apoptosis. Samples were tested in biological replicates (n = 3) and transcript levels were normalized to GAPDH. Cells were harvested in exponential growth phase under normal cell-culturing conditions. Data are expressed as the mean fold change ± SEM of L133 cells relative to HCC 1806 cells. . . Aside fromPPP2R2B (average 2.3-fold change) (p<0.0001), all remaining significant factors showed lower expression. The lowest change was in p21 where the average fold change was 0.02 (p <0.0001). Asterisks in the figure denote the following significance levels: (*) p < 0.05; (**) p < 0.01; (***) p < 0.001; (****) p < 0.0001.

We then wanted to better understand the differences in cell cycle phases by exploring possible connections between TK1 and other cell cycle proteins. RNA levels were quantified using qRT-PCR in HCC 1806 and L133 cells for factors involved in the cell cycle. The factors included were DNA topoisomerase II binding protein 1 (TOPBP1), E2F transcription factor 1 (E2F1), ataxia telangiectasia mutated (ATM), checkpoint kinase 2 (CHEK2), transformation-related protein 53 (TP53), protein phosphatase 2 regulatory subunit Bβ (PPP2R2B), breast cancer gene 1 (BRCA1), TOPBP1 interacting checkpoint and replication regulator (TICRR), clock circadian regulator (CLOCK), retinoblastoma protein (RB1), mouse double minute 2 homolog (MDM2), DNA repair protein RAD52 (RAD52), cyclin Y (CCNY), mitotic arrest deficient 1 like 1 (MAD1L1), and CDK-interacting protein 1 (p21). These factors were selected based on our RNA-seq analysis and the current literature. Samples were tested in biological replicates (n = 3) and were normalized to GAPDH with fold change values comparing L133 cells to HCC 1806 cells. We found that the L133 cells had higher RNA expression for PPP2R2B (average fold change = 2.3) (p<0.0001), while all remaining factors showed lower expression. The lowest change was found for p21 (average fold change = 0.02, p <0.0001). We determined that PPP2R2B and p21 may play a significant role in the cell cycle progression patterns that we observed.

## 5. Examining tumor survival based on effects of TK1 in invasion and apoptosis

Cell cycle progression affects many downstream cellular processes including cell survival and invasion. We hypothesized, based on the results from our cell cycle analysis, that significant differences between HCC 1806 and L133 cells would exist and would contribute to phenotypic differences in the survival and invasion of these two cell types. To test this hypothesis, we quantified levels of apoptosis under differing conditions and measured time to gap closure using a scratch assay.

We first used serum deprivation as a stress condition to compare cell survival between HCC 1806 and L133 cells (Fig 8). We observed that L133 samples (n = 5) had a higher average mean of early apoptotic signs (Annexin V) in both 12-hour samples (p<0.0001) and 30-hour

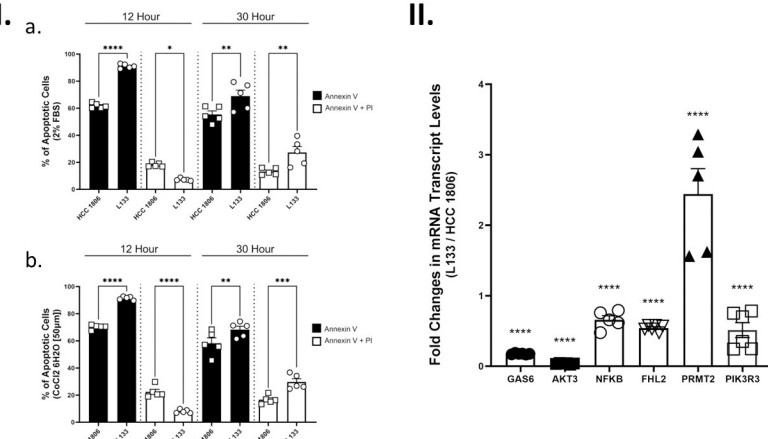

**Fig 8. Investigating the influence of cytosolic TK1 levels in apoptotic response and factors. I.** Apoptosis under serum deprivation and hypoxic conditions for L133 and HCC 1806 cells. Apoptosis was detected by Annexin V$^{FITC}$ and propidium iodide (PI) staining and measured using flow cytometry. **a.** Comparing apoptosis levels in HCC 1806 and L133 samples that were subjected to serum deprived conditions (2% FBS) (n = 5). L133 samples showed higher levels of early apoptotic signs (Annexin V) in both 12- and 30-hour samples. Later apoptotic stages measured with both Annexin V and PI staining showed that HCC 1806 samples stayed relatively stable while L133 samples experienced higher levels of cell death overtime (p< 0.01). **b.** Comparing apoptosis levels in HCC 1806 and L133 samples under hypoxic conditions using $CoCl_2$ 6H2O [50μm] (n = 5). L133 samples showed higher levels of early apoptotic signs (Annexin V) in both 12- and 30-hour samples. Later apoptotic stages measured with both Annexin V and PI staining showed that HCC 1806 samples stayed relatively stable while L133 samples experienced higher levels of cell death overtime (p< 0.001). **II.** Relative transcript levels in HCC 1806 and L133 cells were quantified using qRT-PCR for a subset of factors involved in apoptosis. Samples were tested in biological replicates (n = 3) and transcript levels were normalized to GAPDH. Cells were harvested in exponential growth phase under normal cell-culturing conditions. Data are expressed as the mean fold change ± *SEM* of L133 cells relative to HCC 1806 cells. L133 cells showed lower expression for all factors tested (p <0.0001). Asterisks in the figure denote the following significance levels: (*) p < 0.05; (**) p < 0.01; (***) p < 0.001; (****) p < 0.0001.

samples (p<0.01). Later apoptotic stages, measured with both Annexin V and PI staining, demonstrated that HCC 1806 samples stayed relatively stable while L133 samples experienced higher levels of cell death overtime as shown by the 30-hour samples (n = 5) (p< 0.01).

Hypoxic conditioning was the second stress treatment that we used to compare cell survival between HCC 1806 and L133 cells (Fig 8). Similar to the prior serum deprivation conditions, L133 samples (n = 5) showed higher levels of early apoptotic signs in both 12-hour samples (p<0.0001) and 30-hour samples (p<0.01). The late apoptotic stages again showed similar results to serum deprivation. HCC 1806 samples had relatively consistent Annexin V and PI levels whereas L133 samples experienced higher levels of cell death overtime as shown in the 30-hour samples (n = 5) (p< 0.001). In both contexts, HCC 1806 cells showed increased levels of cell survival when compared to L133 cells.

Considering our apoptotic results, we were very interested in comparing the cell migration of HCC 1806 cells and L133 cells (Fig 9). Once samples had grown to 100% confluency, a scratch in the cell culture wells was made and monitored over a 24-hour period with pictures being taken every hour. We observed that all scratches closed prior to 24 hours. Overall, HCC 1806 cells were significantly more migratory when compared to L133 cells (p <0.001).

Both our apoptotic and migration assays had shown that HCC 1806 cells demonstrated a more pathogenic phenotype that L133 cells. We were curious how TK1 was affecting factors

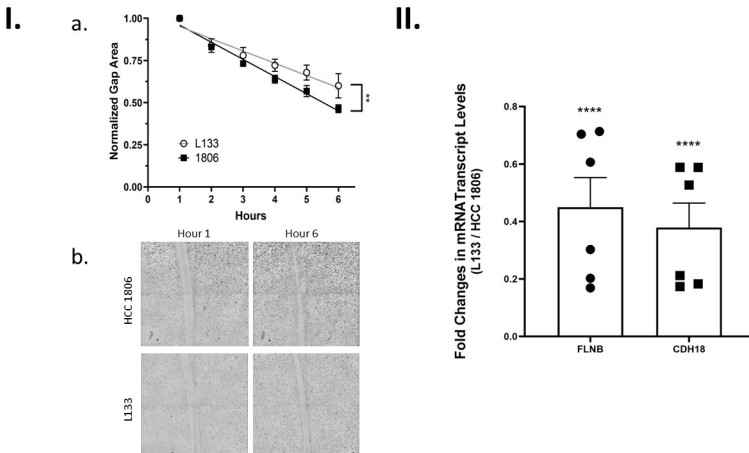

**Fig 9. Exploring the relationship of TK1 on cellular invasion. I.** Wound healing assay comparing migration between L133 and HCC 1806 cells. **a.** The migration ability of HCC 1806 and L133 cells was performed over a 24-hour period in biological replicates (n = 4). Gap area was normalized prior to analysis. HCC 1806 samples showed faster gap closure than L133 samples using a simple linear regression to compare slopes (p<0.01). **b.** Visual representation of HCC 1806 and L133 cell migration at hour 1 and hour 6. **II.** Relative transcript levels in HCC 1806 and L133 cells were quantified using qRT-PCR for a subset of factors involved in apoptosis. Samples were tested in biological replicates (n = 3) and transcript levels were normalized to GAPDH.. Cells were harvested in exponential growth phase under normal cell-culturing conditions. Data are expressed as the mean fold change ± *SEM* of L133 cells relative to HCC 1806 cells.. L133 cells showed lower expression for all factors tested (p <0.0001). Asterisks in the figure denote the following significance levels: (**) p < 0.01 and (****) p < 0.0001.

involved in these pathways and used qRT-PCR to quantify RNA levels of TK1 for factors involved in apoptosis or invasion. We decided to measure expression for these factors in non-stressed conditions to gain a general understanding of how these factors are affected by TK1 expression under normal cell-culturing conditions.

For apoptosis, we decided on comparing expression of GAS6, AKT3, FHL2, MDM2, ATM, TP53, and PRMT2 in HCC 1806 and L133 cells. These factors were highlighted in our RNA-seq analysis and/or are known to be involved in apoptosis. L133 cells showed lower expression for all factors tested except for PRMT2 (p <0.0001 for all factors). The expression for PRMT2, measured with qRT-PCR, almost tripled (average fold change 2.7). GAS6 (avg. fold change 0.18) and AKT3 (avg. fold change 0.03) showed the lowest fold change in expression in L133 cells. The sizeable difference in fold change for PRMT2, GAS6, and AKT3 indicate that TK1 expression may have an impact on these factors and in part explain the pathogenic phenotype we saw in HCC 1806 cells.

We also compared the levels of RNA using qRT-PCR for FLNB and CDH18 in HCC 1806 and L133 cells (Fig 9). Most of the factors tested previously in cell cycle checkpoint or apoptosis qRT-PCR experiments play at least some role in invasion. In addition to those factors, RNA levels were quantified for FLNB and CDH18 because they were significant in our RNA-seq analysis and because of their known role in invasion. The expression for both factors was lower in L133 cells when compared to HCC 1806 cells as shown by the average fold changes: FLNB = 0.45 and CDH18 = 0.38 (p < 0.0001).

The average fold changes of FLNB and CDH18 while significant did not seem dramatic enough to conclude that TK1 expression was influencing cell migration through these factors. Rather, we felt that TK1 may be influencing other factors also involved in invasion that we previously evaluated such as, p21 and PPP2R2B, or factors we did not consider.

## 6. Predicting TK1 interactions in cancer promoting pathways using STRING database

Following qRT-PCR analysis, we wanted to determine the relationship of TK1-interacting factors that we had validated with qRT-PCR. To do so, we used the STRING database to retrieve protein-protein interactions (PPI) to better understand their topological relationship with TK1 in a PPI network (Fig 10). This network showed that 23 of these factors are known to either directly or indirectly interact with TK1 and therefore likely play a role in the observed phenotypic changes between our wild-type and TK1-knockout breast cancer cell lines. Interestingly PRMT2 and GAS6, which we previously supposed were influenced by TK1 expression, showed no connection to TK1. Based on our network and qRT-PCR data, we supposed that TK1 was most likely influencing the phenotype we observed in our study through p21 and AKT3.

## 7. Correlation study of TK1 expression to proteins levels within cancer promoting pathways using BRCA RNA-seq patient samples

Next, we aimed to determine if patient samples correlated with our *in vitro* findings comparing HCC 1806 and L133 cells effects on cell cycle progression, invasion, and apoptosis, (S3–S7 Figs). We performed correlation studies using RNA-seq data from BRCA patients in TCGA (n = 1093) to test for relationships between TK1 expression and known factors within cancer-promoting pathways including invasion, apoptosis, and cell cycle checkpoint using the online TIMER bioinformatic program [31]. We used a Spearman's rank-order correlation to test any correlations that exist between TK1 and known factors that are involved in these pathways. Significant correlations of $p \leq 0.05$ were included to understand general patterns of positive and negative correlations to TK1 expression. Factors for each pathway tested were gathered using peer-reviewed databases PathCard and GeneCard (Table 2) [52, 55].

Cell adhesion factors (n = 118) were studied to identify any potential pattern relating to TK1, and consequently a potential contribution of TK1 to invasion. Overall, we identified a negative relationship (n = 89 out of 118) between TK1 and cell adhesion factors ($p \leq 0.05$). These data imply that overexpression of TK1 potentially inhibits the stability in cell-to-cell adhesions. A majority of the factors most strongly correlated to TK1, either positively or

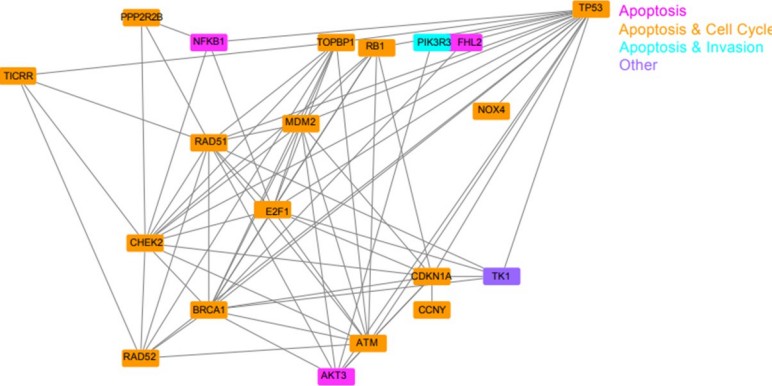

**Fig 10. A TK1 protein-protein interaction network.** Protein nodes are represented by protein symbol and the interactions between these proteins are shown by lines (edges). The legend in the top right corner shows colors associated with the processes of each protein (e.g. apoptosis, cell cycle, invasion, or a combination).

**Table 2. Factors included in each pathway were gathered through PathCard and GeneCard databases.**

**Cell Cycle Checkpoint**

| | | | | | | | | | |
|---|---|---|---|---|---|---|---|---|---|
| ANAPC1 | BUB3 | CDC7 | HUS1 | NBN | PSMA6 | PSMC3 | PSMD8 | RFC5 | UBB |
| ANAPC10 | CCNA1 | CDK1 | KAT5 | NSD2 | PSMA7 | PSMC4 | PSMD9 | RMI1 | UBC |
| ANAPC11 | CCNA2 | CDK2 | MAD1L1 | ORC1 | PSMA8 | PSMC5 | PSME1 | RNF168 | UBE2C |
| ANAPC16 | CCNB1 | CDKN1A | MAD2L1 | ORC2 | PSMB1 | PSMC6 | PSME2 | RNF8 | UBE2D1 |
| ANAPC4 | CCNB2 | CDKN1B | MCM10 | ORC3 | PSMB10 | PSMD1 | PSME3 | RPA1 | UBE2E1 |
| ANAPC5 | CCNE1 | CHEK1 | MCM2 | ORC4 | PSMB11 | PSMD10 | PSME4 | RPA2 | UBE2N |
| ANAPC7 | CCNE2 | CHEK2 | MCM3 | ORC5 | PSMB2 | PSMD11 | PSMF1 | RPA3 | UBE2V2 |
| ATM | CDC16 | CLSPN | MCM4 | ORC6 | PSMB3 | PSMD12 | RAD1 | RPS27A | UIMC1 |
| ATR | CDC20 | COPA | MCM5 | PCBP4 | PSMB4 | PSMD13 | RAD17 | SEM1 | WEE1 |
| ATRIP | CDC23 | DBF4 | MCM6 | PIAS4 | PSMB5 | PSMD14 | RAD50 | SFN | WRN |
| BARD1 | CDC25A | DNA2 | MCM7 | PKMYT1 | PSMB6 | PSMD2 | RAD9A | SUMO1 | YWHAB |
| BLM | CDC25C | EXO1 | MCM8 | PSMA1 | PSMB7 | PSMD3 | RAD9B | TOP3A | YWHAE |
| BRCA1 | CDC26 | GTSE1 | MDC1 | PSMA2 | PSMB8 | PSMD4 | RBBP8 | TOPBP1 | YWHAG |
| BRCC3 | CDC27 | hist1h4c | MDM2 | PSMA3 | PSMB9 | PSMD5 | RFC2 | TP53 | YWHAH |
| BRIP1 | CDC45 | hist2h4a | MDM4 | PSMA4 | PSMC1 | PSMD6 | RFC3 | TP53BP1 | YWHAQ |
| BUB1B | CDC6 | HERC2 | MRE11 | PSMA5 | PSMC2 | PSMD7 | RFC4 | UBA52 | YWHAZ |
| | | | | | | | | | ZNF385A |

**Cell Adhesion**

| | | | | | | | | | |
|---|---|---|---|---|---|---|---|---|---|
| ADAM10 | CCN2 | CDH5 | DIXDC1 | IBSP | ITGB5 | MPZL1 | PLEC | REG4 | TFF1 |
| ADAM9 | CCN3 | CDH6 | DOCK1 | ICAM1 | JUP | MSLN | PLG | SCRIB | TGFB1I1 |
| ADAMTS1 | CD151 | CEACAM1 | DSP | ICAM2 | KLK3 | MTDH | PPFIBP1 | SDC1 | TGFBI |
| AGR2 | CD44 | CEACAM5 | EPCAM | ILK | LAMC1 | MUC1 | PTK2 | SDC4 | TJP1 |
| AMOT | CD63 | CLDN1 | F3 | ITGA4 | LIMS1 | NCAM1 | PTPN12 | SERPINE1 | TJP2 |
| ARHGEF6 | CDCP1 | CLDN3 | FERMT1 | ITGA5 | LPXN | NEDD9 | PTPN14 | SH3PXD2A | TJP3 |
| ARHGEF7 | CDH1 | CTNNA1 | FERMT2 | ITGA6 | MCAM | NEO1 | PTPRA | SIRPA | TLN1 |
| ATP1B1 | CDH11 | CTNNA2 | FERMT3 | ITGAV | MMP13 | OLFM4 | PTPRF | SLK | VCAM1 |
| BCAR1 | CDH17 | CTNNB1 | GIT1 | ITGB1 | MMP14 | PANX1 | PVR | SRCIN1 | VCL |
| BSG | CDH2 | CTNND1 | GIT2 | ITGB2 | MMP2 | PARVA | PXN | ST14 | ZYX |
| CAV1 | CDH3 | CXADR | GJA1 | ITGB3 | MMP3 | PDCD6IP | RAP1A | TACSTD2 | |
| CCN1 | CDH4 | DDR2 | GPNMB | ITGB4 | MMP9 | PLAUR | RAP1B | TFCP2 | |

**Apoptosis**

| | | | | | | | | | |
|---|---|---|---|---|---|---|---|---|---|
| ACVR1 | CAPN11 | CASP9 | FGFR4 | HRAS | ITGA4 | MAPK9 | PDGFRB | PPP2CB | SLTM |
| ACVR1B | CAPN12 | CHEK1 | FLT1 | IGF1R | ITGAV | MDM2 | PDPK1 | PRKCA | TGFBR1 |
| AKT1 | CAPN13 | CHEK2 | FLT4 | IGF2R | ITGB1 | MET | PIK3C2A | PRKCB | TGFBR2 |
| AKT3 | CAPN2 | CHUK | GABRA2 | IKBKB | JUN | MMAB | PIK3C2B | PRKCE | TGFBR3 |
| ANTXR1 | CAPN3 | CSNK2A2 | GABRA3 | IKBKG | KDR | MTOR | PIK3C2G | PRKCG | TP53BP1 |
| ATM | CAPN5 | CSNK2B | GABRA4 | IL13 | KRAS | NFKB1 | PIK3C3 | PRKCH | YWHAE |
| ATR | CAPN6 | CTNNB1 | GABRB1 | IL1A | MAP2K1 | NFKB2 | PIK3CA | PRKCZ | YWHAG |
| BAX | CAPN7 | EGFR | GABRB2 | IL1R1 | MAP2K2 | NGFR | PIK3CD | PTK2 | YWHAH |
| BCL2 | CAPN8 | ERBB2 | GABRD | IL1R2 | MAPK10 | NOS3 | PIK3CG | RELA | YWHAQ |
| BMPR1A | CAPN9 | ERBB3 | GABRE | IL2RA | MAPK12 | NTRK1 | PIK3R1 | RNMT | YWHAZ |
| BMPR1B | CASP10 | ERBB4 | GABRG1 | IL5RA | MAPK13 | NTRK2 | PIK3R4 | RPS6KA2 | |
| CAPN1 | CASP12 | FGFR1 | GABRQ | INSR | MAPK14 | NTRK3 | PIK3R5 | RPS6KA3 | |
| CAPN10 | CASP3 | FGFR2 | GSK3A | ITGA11 | MAPK8 | PDGFRA | PPP2CA | RPS6KA6 | |

**ATM Cell Cycle Checkpoint**

| | | | | | | | | | |
|---|---|---|---|---|---|---|---|---|---|
| ANTXR1 | ATRIP | CCND1 | CDK2 | CHEK2 | MDC1 | NFKB1 | PCNA | RELB | UBA52 |
| ARTN | BLM | CCND2 | CDK4 | CLSPN | MDM2 | NFKB2 | RAD9A | RPS27A | UBB |

*(Continued)*

**Table 2.**  (Continued)

| ATM | BRCA1 | CCND3 | CDKN1A | FANCD2 | MMAB | NFKBIB | RAD9B | SMC1A | UBC |
|---|---|---|---|---|---|---|---|---|---|
| ATR | CCNA2 | CCNE1 | CHEK1 | FANCL | MYC | NFKBIIE | REL | TP53BP1 | USP1 |
|  |  | CDC25A |  | GADD45A |  | RELA |  |  |  |
| **BRCA1 and BRCA2 Homologous Repair** |  |  |  |  |  |  |  |  |  |
| ANTXR1 | ATM | CHEK2 | MDC1 | MSH2 | NTHL1 | POLR2C | POLR2F | POLR2J3 | RAD50 |
| ARTN | BRCA1 | FANCD2 | MMAB | MSH3 | PCNA | POLR2D | POLR2G | POLR2K | RAD51 |
| ATF1 | BRCA2 | H2AFX | MRE11A | MSH6 | POLG2 | POLR2E | POLR2H | POLR2L | TP53BP1 |
|  | BRIP1 |  | MRPL36 |  | POLR2A |  | POLR2J |  | XPC |

negatively, were known to be involved in invasion or metastasis including DIXDC1, NEO1, TJP1, ITGB3, ITGAV, BSG, and MMP9.

Factors generally attributed to apoptosis or cell cycle checkpoints were similarly examined for correlation to TK1 expression. When exploring apoptotic factors for correlation to TK1, 90 out of the 127 factors showed an overall negative correlation ($p \leq 0.05$). These data suggest that the presence of TK1 generally inhibits apoptosis. From the PathCard database we obtained factors within the cell cycle checkpoint pathway (n = 161) and evaluated BRCA RNA-seq patient data for correlations between TK1 and cell cycle checkpoint regulation factors. The general cell cycle checkpoint factors examined for a significant correlation to TK1 demonstrated an overall positive correlation (n = 124) ($p \leq 0.05$), suggesting that TK1 generally does not inhibit cell cycle processes. In addition to testing general apoptotic or cell cycle checkpoint pathways, more specific pathways involved in both cell cycle and apoptotic regulation were also examined for correlations to TK1 including ATM cell cycle checkpoint and BRCA1 and BRCA2 homologous repair. When evaluating these specific pathways, an overall positive correlation was found ($p \leq 0.05$).

## 8. Measuring TK1 levels in breast cancer progression using human breast tissue samples

Based on the patterns we evaluated during *in vitro* testing and validated with patient data, we hypothesized that metastatic breast tissue contained higher levels of TK1 when comparing primary breast cancer tissue to normal breast tissue. To evaluate our hypothesis, an array of breast tissues was stained for TK1 and compared to positive and negative controls. This array included adjacent normal breast (n = 8), infiltrating lobular (n = 7), invasive ductal (n = 39), metastatic ductal (n = 30) and metastatic lobular (n = 6). Our tissue array analysis showed that metastatic ductal and metastatic lobular patient samples displayed higher overall for TK1 expression when compared to normal and primary breast tissue samples ($p < 0.0001$) (Fig 11). Our findings validated our in vitro analyses and additionally suggest that ductal and lobular breast cancer progression is affected by TK1 expression.

## IV. Discussion

In this study, we aimed to explore the relationship(s) between TK1 expression and cancer-promoting pathways involved in breast cancer pathogenesis. We found that increased cell survival and migration was characteristic of HCC 1806 cells and HCC 1806 cells exhibited cell cycle progression patterns indicative of aggressive cell types when compared to the TK1-knockdown L133 cells. Additionally, we selected HCC 1806 cells initially because we were curious if an aggressive primary based breast cancer cell line would be pathogenically altered through

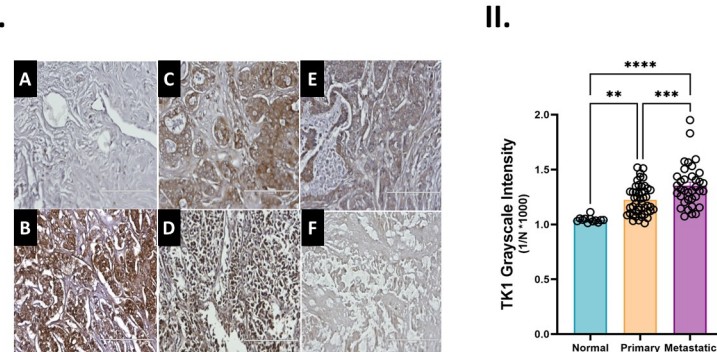

**Fig 11. Elucidating TK1 levels in patient samples. I.** Immunohistochemistry analysis of a breast cancer tissue array. Tissue samples from invasive ductal and infiltrating lobular primary carcinoma patients with matching normal and metastatic samples were stained for TK1 using an HRP conjugated antibody. A) Normal human isotype B) GADPH control C) Metastatic ductal D) Metastatic lobular E) Invasive ductal F) Infiltrating lobular. Assessing associations between gene expression of TK1 and tumor presence through TCGA. **II.** An ANOVA test was computed in PRISM to test differences between means of 3,3′-Diaminobenzidine (DAB) development in tissue samples. Metastatic tissues had a higher presence of TK1 in comparison to normal and primary tumor tissue samples (p value <0.0001).

changes in TK1 expression. Through qRT-PCR and predictive bioinformatics, we found that p21 and AKT3 were among the factors most affected by TK1 expression and the factors most likely through which TK1 was promoting pathogenicity in HCC 1806 cells. A breast cancer tissue array of patient samples verified results we saw in vitro and indicated that ductal and lobular breast cancer progression is influenced by elevated TK1 expression.

Our RNA-seq study comparing HCC 1806 wild-type cells and TK1-knockdown HCC 1806 cells (L133) showed that TK1 was linked to pathogenic pathways including invasion, apoptosis, and cell cycle progression. Prior work has indicated that TK1 is a key player within a network of proteins that affects cellular invasion in breast cancer and tumor progression in both prostate and adrenocortical carcinoma [25–27].

The proper transitioning between G1 to S phase is crucial for DNA damage regulation and oncogenesis [56]. Interestingly, we found that cells in G1 phase were more prevalent within the HCC 1806 population, while the L133 cells showed a higher average number of cells in S phase. The phenotype demonstrated in HCC 1806 cells indicates aggressiveness and when it is compared to the L133 phenotype it suggests that TK1 expression affects the G1-to-S phase transition in these cells [56, 57]. We believed that increased TK1 expression could affect cyclin-dependent kinases (CDK) or peripheral proteins involved in this process such as CDK-interacting protein 1 (p21), pRB, or E2F1 [58].

Although a plethora of possible factors that affect the cell cycle exist, a smaller network of factors associated with TK1 was highlighted that was comprised of 15 factors including PPP2R2B [59], CCNY [57], MAD1L1 [60], TICRR [61], CLOCK [62], p21 [58], pRB [58], E2F1 [58], p53 [6] and RAD52 [63]. Narrowing down the list to these factors enabled us to identify proteins through qRT-PCR that potentially play a strong role in cancer progression, cell cycle checkpoint, cell proliferation, TK1 regulation, and DNA damage. Most factors, 14 out of 15, showed lower expression in the TK1-knockdown cell line L133.

Interestingly, we observed high levels of PPP2R2B in the TK1-knockdown cell line L133 when compared to HCC 1806 cells. Low levels have previously been shown to indicate poor clinical outcome of breast cancer patients [59]. Higher levels of CCNY, which we observed in the HCC 1806 cells, have been shown to be elevated in hepatocellular carcinoma cells (HCC) in comparison to normal tissue; this elevation is linked to increased cell proliferation and

migration, and lower levels of apoptosis; additionally, CCNY has connection to p21 [64]. Interestingly, p21 had the lowest fold change difference of almost zero in our L133 cells. In many cases, p21 has been shown to play a role in the cell cycle by promoting G1 phase arrest [65, 66]. In addition, p21 has been proposed to suppress tumor growth by promoting cell cycle arrest in response to various stimuli [66]. This indicates TK1-knockdown L133 cells are less pathogenic than HCC 1806 cells based on the higher expression of PPP2R2B and lower expression of p21 and CCNY.

Managing transition of cell phases and responding to DNA damage are two major stimulants for apoptosis [65, 67–70]. Our apoptotic analysis showed that L133 cells had higher levels of apoptosis when compared to HCC 1806 cells. This showed that HCC 1806 cells were more resistant to cell stress. We postulate that HCC 1806 cells have increased cell survival due to higher levels of proteins associated with the DNA damage response and increased regulation of the cell cycle when compared to L133 cells [71, 72].

Using qRT-PCR, we quantified the influence of TK1 on apoptotic factors including GAS6 [73], AKT3 [74], PIK3R3 [75], PRMT2 [76, 77], NF-κB [78], and FHL2 [79, 80]. Of note, it is well understood that the behavior of NF-κB is pleiotropic in nature, and while its role in an inflammatory response is a prevalent area of study, it is noted to also play a part in cell cycle regulation and apoptosis [78, 81].

Prior work has shown that PRMT2 expression can have anti- or pro-tumor affects in certain systems. Specifically, decreased expression has been shown to improve the prognosis of breast cancer and HCC patients [76, 82], Alternatively, the presence of PRMT2 has been shown to inhibit NF-κB response and induce apoptosis by blocking the export of IκBα [83]. Additional experiments would be necessary to obtain a more concrete understanding of how increased expression of PRMT2 in L133 cells affects apoptosis.

We were surprised by connections between TK1 and GAS6 as it is involved in Tyro3, Axl and Mer (TAM) receptor signaling and activates downstream signaling through PI3K, ERK, and NF-κB [73]. Through these pathways, GAS6 affects several cellular processes including proliferation, migration, differentiation, adhesion, and apoptosis [73]. In the tumor microenvironment, GAS6 has been shown to have tumor promoting affects by promoting cell proliferation, inhibiting apoptosis, and dampening NK cells anti-tumor effects [69, 84, 85]. In addition, GAS6 overexpression was shown to predict poor prognosis of cancer patients in the clinic [86]. In L133 cells, we did see an increased apoptotic response when compared to HCC 1806 cells in hypoxic and serum starved conditions. We anticipated that the lower levels of GAS6 present in L133 cells may be contributing to the increased apoptosis observed in our system.

AKT3 belongs to the AKT family of three serine/threonine-specific protein kinases, which affect several cellular processes including cell proliferation, migration, metabolism, angiogenesis, and apoptosis [87, 88]. AKT3 specifically has been shown to limit the apoptotic response through NF-κB activation [74, 88]; and by alternating AKT3 activity through mi-RNA silencing breast cancer growth is inhibited [74]. Based on a transcriptomic profiling study, RAS-PI3-K-AKT-NF-κB pathway transcriptionally regulates the expression of BCL2 family and IAP family genes, leading to increased proliferation and apoptosis inhibition [89]. Both BCL2 and IAP families play major roles in committing cells to apoptosis [67, 90]. Interestingly, AKT3 expression was significantly affected by TK1 expression and almost exhibited no expression in the L133 cells. As such, the significant differences we saw in apoptotic populations between the L133 and HCC 1806 cell lines would likely be attributed to lower levels of AKT3 rather than to other apoptotic proteins. The influence of p21-mediated cell cycle arrest and its effects on apoptosis were considered, but not evaluated further in our analysis. It has previously been shown that TK1 elevated expression and activity, such as that seen in HCC 1806 cells, is possible due

to C-terminal regulatory deletions or mutations [91]. In addition, high levels of TK1 have been shown to overcome the inhibitory binding of p21 [91]. We determined that increased TK1 expression in HCC 1806 cells accounted for the cell cycle patterns we observed and explained higher apoptosis levels observed in L133 cells.

In general, it is widely accepted that minimal cellular proliferation and increased levels of apoptosis are key indicators for decreased cellular invasion. The role of TK1 in invasion is not widely understood, but a few recent reports identified TK1 as contributing to MDA-MB-231 cellular invasion [27], as well as cell cycle and pyrimidine nucleoside salvage enzymes [25, 26]. Lower expression of both FLNB and CDH18 have been shown to have anti-invasive effects in mouse embryonic fibroblasts and glioblastoma cells [92, 93]. Surprisingly both FLNB and CDH18 showed approximately half the expression in L133 cells compared to HCC 1806 cells in the current work. As such, we did not feel this change was sufficient to justify a strong relationship between TK1 and FLNB or CDH18.

Recent work showed the epithelial-mesenchymal transition (EMT) associated behavior for cancer metastasis relies on a strong association between downregulation of cyclin-dependent kinases (CDKs) and upregulation of cyclin-kinase inhibitors (CKIs) [94, 95]. Of the CKIs, elevated levels of p21 are critical to facilitate the G1/G0 cell arrest required for EMT to achieve cancer metastasis [95, 96]. In addition, p21 has been shown to rescue the invasive ability of nhr-67-deficient anchor cells by inducing G1/G0 arrest and restoring formation of invadopodia and matrix metalloproteinases (MMPs) [96]. This suggests that p21 expression is aiding the migration of HCC 1806 cells through cell cycle arrest.

Although detecting GAS6 with qRT-PCR caused us to anticipate some connection to TK1, the protein-protein interaction network showed no direct interactions between TK1 and GAS6, FLNB, PRMT2, CDH18, MAD1L1, or CLOCK. Additional experimentation is required to determine whether such a direct interaction exists in this system. Some of the interactions between TK1 and this protein-protein interaction network are well established such as those with E2F1, CDKN1A (p21), and RB1; but surprisingly, there was a direct link to p53, which we did not anticipate. While we acknowledge that our interaction network contains only a subset of factors, there is a clear connection to TK1, AKT3, and p21 through the DNA damage response [2, 97, 98]. This connection includes interactions involving RAD51, RAD52, CHEK2, BRCA1, RB1, E2F1, p53, PIK3R3, and ATM.

Based on the evidence provided through our study, HCC 1806 cells endogenously express high levels of TK1 and this expression contributes to the pathogenic phenotype of HCC 1806 cells. The increased levels of TK1 are likely caused by dysregulation at its promoter site and/or a mutated C-terminus—which prevents ubiquitin-dependent degradation of TK1. Either of these events would potentially allow TK1 to be present during all phases of the cell cycle [91]. It is possible that the increased presence of TK1 in our system randomly affected other intracellular factors such as p21 and AKT3. However, the dramatic effect that TK1 expression had on altering expression of specific factors indicates the expression patterns we observed were not happening by chance. Downstream of TK1 dysregulation, we think it is highly likely that increased levels of TK1 contribute to a negative-feedback loop affecting protein expression for specific factors—including p21 and AKT3. Considering our data, we believe that this negative feedback loop is comprised of DNA damage response and cell cycle regulation proteins. Accounting for the different factors affected by TK1 expression and how they are related, a negative feedback loop would be plausible but would need further experimentation to confirm. As such, analysis following a recovery of TK1 expression in L133 cells would be an important avenue to explore.

The tumor microenvironment is highly complex and constantly experiencing cellular stress due to factors such as inflammation, hypoxia, and glucose depletion; as a result, DNA damage

is present at all stages of tumor development and progression [99, 100]. Therefore, tumor cells are constantly being challenged to survive through added mutations and/or DNA damage repair mechanisms. Higher TK1 levels provide cells with an increased ability to combat internal and external stresses that cause DNA damage. The phenotype observed in HCC 1806 cells compared to L133 cells in cell cycle analysis, apoptosis assays, and scratch assay suggest that TK1 expression promotes HCC 1806 pathogenicity. Whether the differences in cell cycle, apoptosis, and growth are directly affected by TK1 in HCC 1806 cells will need further elucidation. A limitation in our study is not evaluating the expression of qRT-PCR factors in stressed or pharmacologically-induced conditions for cell cycle, apoptosis, or cellular migration/invasion. We plan to include those evaluations in future studies. We also acknowledge our study is limited to comparisons in HCC 1806 and L133 cells and would be enhanced by extending our study to other breast cancer cell models. We also plan to explore gene silencing of TK1 using pharmacological and translational based methods and the effects of TK1 overexpression. However, based on our current understanding we expect that our experimental observations occur most directly through p21 and AKT/PIK3R3 signaling. We also expect that the phenotypic changes conferred by overexpression of TK1 in these cells is at least somewhat independent of p53. Further evaluation of the role that cytosolic TK1 plays in the tumor environment and its influence on breast cancer progression is essential in understanding its potential as both a cancer biomarker and a potential therapeutic target.

## V. Conclusion

This study provides strong evidence that TK1 directly contributes to the increased pathogenicity of HCC 1806 cells by p21- and AKT3-mediated mechanisms that include promoting cell cycle arrest, cellular migration, and cellular survival. We expect that these findings will be incorporated into future mechanistic and/or therapeutic studies to improve patient outcomes and survival.

## Supporting information

**S1 Fig. Evaluating TK1 expression between normal and cancer patients.** RNA TK1 differential expression was examined between adjacent normal and tumor tissue. RNA-seq data was extracted from The Cancer Genome Atlas (TCGA) for 39 different cancer types. TK1 RNA expression was significantly higher in cancer tissue when compared to adjacent normal samples for 20 of the 39 cancer types evaluated. Wilcoxon test was used to test significance and is shown by the number of stars (**: p-value <0.01; ***: p-value <0.001). Columns in gray denote when normal data are available.
(TIF)

**S2 Fig. TK1 regulation. Promoter methylation level of TK1 in BRCA patients.** The data shown is described using a box and whisker plot and shows the average β value. The beta value indicates level of DNA methylation ranging from 0 (unmethylated) to 1 (fully methylated). The difference between the tumor and normal samples was estimated by a student's t-test in which unequal variance was considered. Normal patient samples showed a higher level of methylation in the TK1 promoter when compared to primary tumor samples (p = 1.11 x 10–16).
(TIF)

**S3 Fig. RNA-seq data evaluated in BRCA patients (n = 1095) for correlations between cell cycle checkpoint factors and TK1.** Significant Spearman correlations (p ≤ 0.05) are shown for positive (n = 124) and negative (n = 37) coefficients. Stronger correlations extend above or

below two standard deviations that are denoted by horizontal asymptotes at 0.408 and -0.159.
(TIF)

**S4 Fig. RNA-seq data evaluated in BRCA patients (n = 1095) for correlations between apoptotic factors and TK1.** Significant Spearman correlations (p ≤ 0.05) are shown for positive (n = 37) and negative (n = 90) coefficients. Stronger correlations extend above or below two standard deviations that are denoted by horizontal asymptotes at 0.24 and -0.20.
(TIF)

**S5 Fig. RNA-seq data evaluated in BRCA patients (n = 1095) for correlations between cell adhesion factors and TK1.** Significant Spearman correlations (p ≤ 0.05) are shown for positive (n = 29) and negative (n = 89) coefficients. Stronger correlations extend above or below two standard deviations that are denoted by horizontal asymptotes at 0.16 and -0.19.
(TIF)

**S6 Fig. RNA-seq data evaluated in BRCA patients (n = 1095) for correlations between ATM checkpoint factors and TK1.** Significant Spearman correlations (p ≤ 0.05) are shown for positive (n = 28) and negative (n = 15) coefficients. Stronger correlations extend above or below two standard deviations that are denoted by horizontal asymptotes at 0.41 and -0.20.
(TIF)

**S7 Fig. RNA-seq data evaluated in BRCA patients (n = 1095) for correlations between BRCA1 and BRCA2 checkpoint factors and TK1.** Significant Spearman correlations (p ≤ 0.05) are shown for positive (n = 26) and negative (n = 9) coefficients. Stronger correlations extend above or below two standard deviations that are denoted by horizontal asymptotes at 0.35 and -0.19.
(TIF)

**S1 Raw images. Unedited western blot images for Figs 2 and 3.** This file is submitted as a PDF.
(PDF)

**S1 Data. Data used for analyses.** Data in experiments and analyses are provided for in this Excel file.
(XLSX)

**S2 Data. RNA-seq files comparing HCC 1806 and L133 cells.** RNA-seq was not funding for federally. Excel files shared through LC Sciences are made available in the zip file.
(ZIP)

## Acknowledgments

We would like to thank Zac E. Ence and Stephen R. Piccolo and recognize their contribution to data analysis, interpretation of data, and visualization for the comparison of metastatic and primary cancer cell line TK1 transcript levels. We also would like to thank Brigham Young University and Thunder Biotech for their support in the completion of this project and their generosity in allowing us to use laboratory space and equipment.

## Author Contributions

**Conceptualization:** Eliza E. Bitter, Rachel Brog, Tim Phares, Michelle Townsend, Kim L. O'Neill.

**Data curation:** Carolyn I. Allen, Rachel I. Erickson, Rachel M. Morris, Toni Mortimer, Audrey Meade.

**Formal analysis:** Eliza E. Bitter, Rachel Brog, Tim Phares, Michelle Townsend, Brett E. Pickett, Kim L. O'Neill.

**Methodology:** Jonathan Skidmore, Carolyn I. Allen, Rachel I. Erickson, Rachel M. Morris, Toni Mortimer, Audrey Meade, Kim L. O'Neill.

**Project administration:** Kim L. O'Neill.

**Resources:** Kim L. O'Neill.

**Writing – original draft:** Eliza E. Bitter, Brett E. Pickett, Kim L. O'Neill.

**Writing – review & editing:** Eliza E. Bitter, Jonathan Skidmore, Rachel Brog, Tim Phares, Michelle Townsend, Brett E. Pickett, Kim L. O'Neill.

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
