## [Decision Letter · Decision Letter 0]

13 Jul 2023

PONE-D-23-07912PLOS ONE TK1 expression influences pathogenicity by cell cycle progression, cellular migration, and cellular survival in HCC1806 Breast Cancer CellsPLOS ONE

Dear Dr. O’Neill,

Thank you for submitting your manuscript to PLOS ONE. After careful consideration, we feel that it has merit but does not fully meet PLOS ONE’s publication criteria as it currently stands. Specifically, two of the reviewers have raised some concerns that should be addressed before we can move forward. Therefore, we invite you to submit a revised version of the manuscript that addresses the points raised during the review process.

We look forward to receiving your revised manuscript.

Kind regards,

Zhiming Li, Ph.D.

Academic Editor

PLOS ONE

Journal Requirements:

Reviewers' comments:

Reviewer's Responses to Questions

**Comments to the Author**

1. Is the manuscript technically sound, and do the data support the conclusions?

Reviewer #1: Yes

Reviewer #2: Yes

Reviewer #3: Yes

2. Has the statistical analysis been performed appropriately and rigorously? 

Reviewer #1: Yes

Reviewer #2: Yes

Reviewer #3: Yes

3. Have the authors made all data underlying the findings in their manuscript fully available?

Reviewer #1: Yes

Reviewer #2: Yes

Reviewer #3: Yes

4. Is the manuscript presented in an intelligible fashion and written in standard English?

Reviewer #1: Yes

Reviewer #2: Yes

Reviewer #3: Yes

5. Review Comments to the Author

Reviewer #1: This was a well designed study, showed cell cycle progression, apoptosis, and invasion as potential pathogenic pathways affected by elevated TK1 expression; and provides evidence that TK1 directly contributes to the increased

pathogenicity of HCC 1806 cells by p21- and AKT3-mediated mechanisms that include promoting cell cycle arrest, cellular migration, and cellular survival. I do recommend the author shorted the manuscript and make the paper more readable.

Reviewer #2: The manuscript entitled “TK1 expression influences pathogenicity by cell cycle progression, cellular

migration, and cellular survival in HCC1806 Breast Cancer Cells” by Bitter et al. is a well conducted study showing that the elevated expression of Thymidine Kinase 1 (TK1) is associated with more aggressive tumor grades. Specifically, the authors focused their attention on breast cancer demonstrating that TK1 is associated with increased pathogenicity when comparing the breast cancer cell line HCC1806 with the L133 that is knock-out for TK1 expression.

Its relevance to cancer is supported by the fact that TK1 expression is higher in metastatic tissues compared to primary ones suggested that could be involved in tumor progression. However some concerns have to be raised to the present version of the manuscript.

Major point:

1) The entire paper is based on two cancer cell models. It should be better to extend these findings to other cell models or with multiple approaches (shRNA, overexpression)

2) There is a way to pharmacologically inhibit or block TK1 activity?

3) Pag. 27. Please specify the type of tumors evaluated for TK1 expression.

4) Figure 2, I. Please show the western blot of TK1 protein expression. In the graph specify the difference between the white and gray columns.

5) Figure 3, II, a. Please indicate which PCR is shown and for which samples (from 1 to 6)

6) Figure 3, III and Figure 7, I. Please explain why L133 that has a lower TK1 content has a higher frequency of division and a higher S phase.

7) Figure 7, II and figure 8, II. Please specify the Fold increase of what is shown in the graph.

8) Figure 8, I, a and b. Please specify the difference between the two graphs, not only in the text.

9) Figure 9, II. Please explain how has been obtained the graph.

Minor points:

10) Pag. 11. Correct the sentence: “Solid Tissue Normal” with “solid normal tissue”

11) Pag. 11 and pag. 27. In Supporting Information 1: correct the word: “Wlicoxon test” with “Wilcoxon test”

12) Pag. 28. Fix the following phrase: “As we categorized these cell lines as either “metastatic” or “primary” based on available literature. We confirmed metastatic breast cancer cell lines contained higher levels of TK1 than primary breast cancer cell”

13) Pag. 29. Correct the sentence: “Next wanted to …” with “Next we wanted to …”

14) Pag. 29. Correct the sentence: “primary breast cell line” with “primary breast cancer cell line”

15) Pag. 42. Correct the sentence: “In many cases, p21 has be …” with “In many cases, p21 has been …”

Reviewer #3: This study investigates the effects of TK1 expression on pathogenic phenotypes of HCC 1806 breast cancer cells. The study provides a novel and interesting data set generated by transcriptome analysis.

1. Overall, written styles need to be improved. Specifically, the method section could be more concise, and the manuscript should have fewer number references by selecting only the best reference for statements.

2. Concerns/questions are needed to be addressed.

2.1. In method section (8), TK1 quantification of qPCR, there should be primer sequences of TK1 gene.

2.2. Why did the authors select HCC 1806 cell lines for this study? Figure 2 shows no difference in TK1 protein expression levels in different cell lines tested, MCF-7, MDA-MB-231, T47D, and JIMT-1. Is there any specific reason for choosing an HCC180 cell line?

2.3. In Figure 2 (II), which cell lines were categorized in the metastatic or primary cells tested for TK1 protein expression?

2.4. Figure 3 and in the method (title no. 9), how did the authors calculate frequency (as division per day)?

2.5. In RNA-seq analysis (Gene ontology, Figure 4-5-6), Go term and KEGG enrichment analysis identify cell adhesion genes as the top significant pathway, but it is not quite clear how to obtain the selected genes involving cell apoptosis and cell cycle from these analyses.

2.6. There is no explanation of cell culture conditions for qPCR analysis in Figure 8 (II), hypoxia or serum deprivation, 12 or 30 hours.

2.7. Which time point of cell culture was made for qPCR analysis in Figure 9 (II)?

2.8. According to the results of cell cycle and apoptosis assays (Figure 7 and 8), the WT HCC 1806 cell culture, which showed a higher no. of cells in G1 phase and higher expression levels of anti-tumor P21 (which possibly induces G1 cell cycle arrest), has a lower percentage of apoptotic cells. Should there be a correlation between P21-mediated cell cycle arrest and apoptosis levels in the culture? Is there any more explanation?

6. PLOS authors have the option to publish the peer review history of their article (what does this mean?). If published, this will include your full peer review and any attached files.

Reviewer #1: **Yes: **Youxin Ji MD

Reviewer #2: No

Reviewer #3: No

---

## [Author Response · Author response to Decision Letter 0]

15 Sep 2023

Dear Academic Editor and Reviewers:

We sincerely appreciate the feedback we received regarding our manuscript and are submitting the following rebuttal letter in hopes to be approved for resubmission to PLOS ONE.

The research article enclosed has been edited based on the comments we received. Below, you will find each point raised and how each was addressed by our authors. Our manuscript is an original work and an honest representation of the repeatable results observed during the study. We feel that the study performed fits within the aims and scope of the journal, especially with the improvements made based on initial review, and would like to resubmit for further review.

I. Editor Comments

1. Please ensure that your manuscript meets PLOS ONE's style requirements, including those for file naming. The PLOS ONE style templates can be found at https://journals.plos.org/plosone/s/file?id=wjVg/PLOSOne_formatting_sample_mai n_body.pdf and https://journals.plos.org/plosone/s/file?id=ba62/PLOSOne_formatting_sample_title _authors_affiliations.pdf

We apologize that there were issues with this first requirement following our initial submission. We have reviewed the style requirements and made changes based on the style templates provided for by PLOS ONE.

2. PLOS ONE now requires that authors provide the original uncropped and unadjusted images underlying all blot or gel results reported in a submission’s figures or Supporting Information files. This policy and the journal’s other requirements for blot/gel reporting and figure preparation are described in detail at https://journals.plos.org/plosone/s/figures#loc-blot-and-gel-reporting-requirements and https://journals.plos.org/plosone/s/figures#loc-preparing-figures-from-image- files. When you submit your revised manuscript, please ensure that your figures adhere fully to these guidelines and provide the original underlying images for all blot or gel data reported in your submission. See the following link for instructions

on providing the original image data: https://journals.plos.org/plosone/s/figures#loc-original-images-for-blots-and-gels.

We apologize that the requested images were not available in our first submission and have provided the uncropped and unadjusted images for the Western Blots in our manuscript. The images are provided in our supporting information. We additionally noted in our cover letter the location of these images.

Please see cover letter for changes in our Data Availability statement.

All references were crossed checked using a retracted article database (https://retractionwatch.com/retraction-watch-database-user-guide/) we did not note any articles that have been retracted that our cited in our manuscript.

II. Reviewer #1

This was a well-designed study, showed cell cycle progression, apoptosis, and invasion as potential pathogenic pathways affected by elevated TK1 expression; and provides evidence that TK1 directly contributes to the increased

pathogenicity of HCC 1806 cells by p21- and AKT3-mediated mechanisms that include promoting cell cycle arrest, cellular migration, and cellular survival. I do recommend the author shortened the manuscript and make the paper more readable.

Please note the edits made in the revised copy of our manuscript to address the readability and length of our study.

III. Reviewer #2

 2

The manuscript entitled “TK1 expression influences pathogenicity by cell cycle progression, cellular migration, and cellular survival in HCC1806 Breast Cancer Cells” by Bitter et al. is a well conducted study showing that the elevated expression of Thymidine Kinase 1 (TK1) is associated with more aggressive tumor grades. Specifically, the authors focused their attention on breast cancer demonstrating that TK1 is associated with increased pathogenicity when comparing the breast cancer cell line HCC1806 with the L133 that is knock-out for TK1 expression. Its relevance to cancer is supported by the fact that TK1 expression is higher in metastatic tissues compared to primary ones suggested that could be involved in tumor progression. However, some concerns have to be raised to the present version of the manuscript.

Major point:

1) The entire paper is based on two cancer cell models. It should be better to extend these findings to other cell models or with multiple approaches (shRNA, overexpression).

The point raised is very insightful and valid. At this time, we will not be able to provide these results as they are being included in future/current studies. However, there importance and relevance to our manuscript are addressed in the “Discussion” section of our paper:

We also acknowledge our study is limited to comparisons in HCC 1806 and L133 cells and would be enhanced by extending our study to other breast cancer cell models. In addition to expanding our study to other breast cancer cell models, we plan to explore gene silencing of TK1 using pharmacological and translational based methods and the effects of TK1 overexpression.

2) There is a way to pharmacologically inhibit or block TK1 activity?

Yes, these methods are available. Please see comments made for major point 1.

3) Pag. 27. Please specify the type of tumors evaluated for TK1 expression.

To avoid confusion in the manuscript, we made the following change: “Normal patient samples showed a higher level of methylation in the TK1 promoter when compared to primary BRCA tumor samples (p=1.11 x 10-16) (S2 Fig).”

4) Figure 2, I. Please show the western blot of TK1 protein expression. In the graph specify the difference between the white and gray columns.

A representative Western Blot image was provided for in Figure 2 and a legend was added to clarify white and gray columns for the reader.

5) Figure 3, II, a. Please indicate which PCR is shown and for which samples (from 1 to 6). Changes were made in the figure as well as in the description of the figure to address confusion for the PCR samples shown.

6) Figure 3, III and Figure 7, I. Please explain why L133 that has a lower TK1 content has a higher frequency of division and a higher S phase.

We are grateful that the reviewer had comments regarding TK1 expression in L133 cells and HCC 1806 cells and the comparison between frequency of division and S phase. Following further investigation of our original data, we found that the analysis for HCC 1806 and L133 cells had been switched for frequency of division. We are unsure how this error was originally made and again feel very grateful for Reviewer #2 in raising this as a point of concern. While there is a significant difference still between the two cell lines in division, we feel it isn’t biologically relevant to change the results of our study. This “biological relevance” was already addressed in our first submitted manuscript.

7) Figure 7, II and figure 8, II. Please specify the Fold increase of what is shown in the graph.

3

In figure 7, II and figure 8, II the y axis label was changed to indicate the fold change is transcript fold change and that it is comparing L133 cells to HCC 1806.

8) Figure 8, I, a and b. Please specify the difference between the two graphs, not only in the text.

The y axis label for figure 8, I, a and b was changed to indicate the apoptotic population measured.

9) Figure 9, II. Please explain how has been obtained the graph.

In figure 9, II the y axis label was changed to indicate the fold increase. In addition, the caption was edited and more details were included in the manuscript to indicate how cells were harvested for this analysis.

Minor points:

10) Pag. 11. Correct the sentence: “Solid Tissue Normal” with “solid normal tissue”

This change was made in the revised manuscript.

11) Pag. 11 and pag. 27. In Supporting Information 1: correct the word: “Wlicoxon test” with “Wilcoxon test”

This change was made in the revised manuscript.

12) Pag. 28. Fix the following phrase: “As we categorized these cell lines as either “metastatic” or “primary” based on available literature. We confirmed metastatic breast cancer cell lines contained higher levels of TK1 than primary breast cancer cell”

This phrase was changed to: “These cell lines were categorized as either “metastatic” or “primary” based on available literature. Our Western Blots indicated that metastatic breast cancer cell lines contained higher levels of TK1 when compared to primary breast cancer cell lines.”

13) Pag. 29. Correct the sentence: “Next wanted to ...” with “Next we wanted to ...”

This change was made in the revised manuscript.

14) Pag. 29. Correct the sentence: “primary breast cell line” with “primary breast cancer cell line”

This change was made in the revised manuscript.

15) Pag. 42. Correct the sentence: “In many cases, p21 has be ...” with “In many cases, p21 has been ...”

This change was made in the revised manuscript.

IV. Reviewer #3:

This study investigates the effects of TK1 expression on pathogenic phenotypes of HCC 1806 breast cancer cells. The study provides a novel and interesting data set generated by transcriptome analysis.

1. Overall, written styles need to be improved. Specifically, the method section could be more concise, and the manuscript should have fewer number references by selecting only the best reference for statements.

Edits were made in the revised copy of our manuscript to address the length of our study. The references were also reviewed, but we felt that all were worthwhile to keep in the manuscript. 2. Concerns/questions are needed to be addressed.

2.1. In method section (8), TK1 quantification of qPCR, there should be primer sequences of TK1 gene.

Primer sequences were added for TK1 quantification in the method section of our manuscript.

4

2.2. Why did the authors select HCC 1806 cell lines for this study? Figure 2 shows no difference in TK1 protein expression levels in different cell lines tested, MCF-7, MDA-MB-231, T47D, and JIMT-1. Is there any specific reason for choosing an HCC180 cell line?

Figure 2 was edited to clarify differences in metastatic and primary based breast cancer cell lines. Additionally, we added the following phrases in our manuscript to explain our reasoning for HCC 1806 selection:

“Our selection of HCC 1806 cells was focused on their innate aggressive nature as a primary based breast cancer cell line, and we were curious if that nature could be pathogenically altered through changes in TK1 expression.”

“Additionally, we selected HCC 1806 cells initially because we were curious if an aggressive primary based breast cancer cell line would be pathogenically altered through changes in TK1 expression.”

2.3. In Figure 2 (II), which cell lines were categorized in the metastatic or primary cells tested for TK1 protein expression?

Figure 2 and the caption were both edited to address this point.

2.4. Figure 3 and in the method (title no. 9), how did the authors calculate frequency (as division per day)?

Figure 3 caption was edited, and the following was added to the method section (title no. 9): “Individual standard curves for HCC 1806 and L133 cells were then used to determine the cell count based on color development readings. From here, we were able to calculate the average cell division per day for both cell types.”

2.5. In RNA-seq analysis (Gene ontology, Figure 4-5-6), Go term and KEGG enrichment analysis identify cell adhesion genes as the top significant pathway, but it is not quite clear how to obtain the selected genes involving cell apoptosis and cell cycle from these analyses.

This point of concern is valid and was addressed in the revised manuscript through the following:

The Gene ontology (GO) analysis of differentially expressed genes identified cell adhesion as a significant pathway linked to TK1 expression. Similarly, KEGG enrichment analysis identified cell adhesion molecules as a significant pathway. GO analysis also identified the apoptosis process as a significant GO term, and additional terms relating to the cell cycle as significant GO terms (Fig 4). Some of those cell cycle terms include variations of regulation of transcription, positive regulation of cell population proliferation, and positive regulation of gene expression.

2.6. There is no explanation of cell culture conditions for qPCR analysis in Figure 8 (II), hypoxia or serum deprivation, 12 or 30 hours.

Clarification was made for cell culturing conditions in figure 8 caption and in the manuscript. We hoped to understand the basal level of influence TK1 had on HCC 1806 and L133 cells. As such, qPCR analysis was made under non-stressed cellular conditions for apoptotic factors. We understand that this is a limitation of our study, which we address in the discussion, and plan to evaluate these factors more thoroughly in future planned studies.

2.7. Which time point of cell culture was made for qPCR analysis in Figure 9 (II)?

Clarification was made for cell culturing conditions in figure 9 caption and in the manuscript. We hoped to understand the basal level of influence TK1 had on HCC 1806 and L133 cells. As such, qPCR analysis was made under non-stressed cellular conditions for cellular migration and

5

were not harvested at a particular timepoint following the scratch assay. We understand that this is a limitation of our study, which we address in the discussion, and plan to evaluate these factors more thoroughly in future planned studies.

2.8. According to the results of cell cycle and apoptosis assays (Figure 7 and 8), the WT HCC 1806 cell culture, which showed a higher no. of cells in G1 phase and higher expression levels of anti-tumor P21 (which possibly induces G1 cell cycle arrest), has a lower percentage of apoptotic cells. Should there be a correlation between P21-mediated cell cycle arrest and apoptosis levels in the culture? Is there any more explanation?

The following addition was added to our manuscript to address this point:

The influence of p21-mediated cell cycle arrest and its effects on apoptosis were considered, but not evaluated further in our analysis. It has previously been shown that TK1 elevated expression and activity, such as that seen in HCC 1806 cells, is possible due to C-terminal regulatory deletions or mutations. In addition, high levels of TK1 have been shown to overcome the inhibitory binding of p21. We determined that increased TK1 expression in HCC 1806 cells accounted for the cell cycle patterns we observed and explained higher apoptosis levels observed in L133 cells.

Best Regards,

Kim O’Neill

Professor

Brigham Young University

---

## [Decision Letter · Decision Letter 1]

6 Oct 2023

PLOS ONE TK1 expression influences pathogenicity by cell cycle progression, cellular migration, and cellular survival in HCC1806 Breast Cancer Cells

PONE-D-23-07912R1

Dear Dr. O’Neill,

We’re pleased to inform you that your manuscript has been judged scientifically suitable for publication and will be formally accepted for publication once it meets all outstanding technical requirements.

Kind regards,

Zhiming Li, Ph.D.

Academic Editor

PLOS ONE

Additional Editor Comments (optional):

Reviewers' comments:

Reviewer's Responses to Questions

**Comments to the Author**

1. If the authors have adequately addressed your comments raised in a previous round of review and you feel that this manuscript is now acceptable for publication, you may indicate that here to bypass the “Comments to the Author” section, enter your conflict of interest statement in the “Confidential to Editor” section, and submit your "Accept" recommendation.

Reviewer #3: All comments have been addressed

2. Is the manuscript technically sound, and do the data support the conclusions?

Reviewer #3: Yes

3. Has the statistical analysis been performed appropriately and rigorously? 

Reviewer #3: Yes

4. Have the authors made all data underlying the findings in their manuscript fully available?

Reviewer #3: Yes

5. Is the manuscript presented in an intelligible fashion and written in standard English?

Reviewer #3: Yes

6. Review Comments to the Author

Reviewer #3: This work is well designed and a revised manuscript is more readable. All comments in the previous round have been clearly addressed.

7. PLOS authors have the option to publish the peer review history of their article (what does this mean?). If published, this will include your full peer review and any attached files.

Reviewer #3: No

---

## [Editor Report · Acceptance letter]

17 Nov 2023

PONE-D-23-07912R1 

TK1 expression influences pathogenicity by cell cycle progression, cellular migration, and cellular survival in HCC 1806 breast cancer cells 

Dear Dr. O’Neill:

I'm pleased to inform you that your manuscript has been deemed suitable for publication in PLOS ONE. Congratulations! Your manuscript is now with our production department. 

Kind regards, 

on behalf of

Dr. Zhiming Li 

Academic Editor

PLOS ONE